# The formation of the 'footprint of death' as a mechanism for generating large substrate-bound extracellular vesicles that mark the site of cell death

Stephanie F. Rutter [1,2], Taeyoung Kang [1,2], Gemma F. Ryan[1,2], Bo Shi [1,2], Caitlin L. Vella[1,2,3], Pradeep Rajasekhar [4,5], Sean W. Cutter[1], Amy L. Hodge[1], Dilara C. Ozkocak[1,2], Ching-Seng Ang[6], Julian Ratcliffe [7], Katrina J. Binger[1], Pamali Foneska[1,2], Suresh Mathivanan [1,2], Niall D. Geoghegan [4,5], Kelly L. Rogers [4,5], Michael F. Olson [8], Georgia K. Atkin-Smith[4,5,9] ✉ & Ivan K. H. Poon [1,2,9] ✉

Apoptotic cells communicate to phagocytic cells through releasing soluble factors and apoptotic cell-derived extracellular vesicles. However, whether there are additional factors that remain attached at the site of cell death to signal to phagocytic cells is currently unknown. Here we show that apoptotic cell retraction generates a membrane-encased, F-actin-rich 'footprint' tightly anchored to the substrate that marks the site of cell death, coined 'the FOot-print Of Death' or FOOD. Formation of FOOD is observed frequently across many different cell types, apoptotic stimuli and surface composition. Mechanistically, FOOD formation is regulated by the protein kinase ROCK1. 3D time-lapse microscopy studies revealed that FOOD vesicularises into distinct large extracellular vesicles. These extracellular vesicles expose the 'eat-me' signal phosphatidylserine and can function to 'flag' the site of cell death to neighbouring phagocytes for efferocytosis. Under a viral infection setting, FOOD can harbour viral proteins and virions, and propagate infection to healthy cells. Together, this study has revealed another route of apoptotic cell-phagocyte communication.

Following commitment to cell death, apoptotic cells initiate important communication with neighbouring cells to facilitate phagocyte recruitment for cell clearance (i.e. efferocytosis) as well as wound healing and anti-inflammatory responses[1-3]. This communication is mediated by the release of soluble 'find-me' and 'good-bye' signals, the exposure of 'eat-me' signals on the outer membrane leaflet, and the release of apoptotic cell-derived extracellular vesicles (ApoEVs) in particular, apoptotic bodies (ApoBDs)[1,3-7]. ApoBDs are a subset of large

[1]Department of Biochemistry and Chemistry, La Trobe Institute for Molecular Science, La Trobe University, Melbourne, VIC, Australia. [2]Research Centre for Extracellular Vesicles, La Trobe University, Melbourne, VIC, Australia. [3]Centre for Cardiovascular Biology and Disease Research, La Trobe Institute for Molecular Science, La Trobe University, Bundoora, VIC, Australia. [4]The Walter and Eliza Hall Institute of Medial Research, Parkville, VIC, Australia. [5]University of Melbourne, Melbourne, VIC, Australia. [6]The Bio21 Institute of Molecular Science and Biotechnology Institute, University of Melbourne, Parkville, VIC, Australia. [7]Bioimaging Platform, La Trobe University, Melbourne, VIC, Australia. [8]Department of Chemistry and Biology, Toronto Metropolitan University, Toronto, ON, Canada. [9]These authors contributed equally: Georgia K. Atkin-Smith, Ivan K. H. Poon. ✉e-mail: atkinsmith.g@wehi.edu.au; i.poon@latrobe.edu.au

ApoEVs, typically 1–5 μm in diameter that can harbour biomolecules including DNA, RNA, and proteins, and pathogens[8–14]. Following interaction with recipient cells, ApoBDs can influence the local tissue microenvironment by promoting stem cell proliferation, antigen presentation, and propagation of viral infections[2,8,10,12,15,16]. In addition to interacting with cells at the site of cell death, ApoBDs can disperse through the circulation and thus may exert functional effects at distal sites[12,17,18]. Notably, the release of small EV subsets such as exosome-like ApoEVs by apoptotic cells has also been reported to aid processes such as cell clearance and tissue regeneration[19–21]. Given the broad importance of ApoEVs under both homeostatic and disease settings, understanding the molecular mechanisms responsible for ApoEV formation is vital.

The formation of ApoBDs is regulated by stepwise morphological changes during apoptosis, a process known as apoptotic cell disassembly[22,23]. During the early stages of apoptosis, caspase-activated Rho-associated kinase 1 (ROCK1) phosphorylates myosin light chain, leading to the actomyosin contraction necessary for apoptotic membrane blebbing[24–27]. Notably, for adherent cell types, this contractile force also drives the cell contraction necessary for cell detachment from the substratum[28]. After plasma membrane blebbing, the apoptotic cell can generate thin membrane protrusions called apoptopodia and beaded-apoptopodia that actively radiate from the apoptotic cells to facilitate cell fragmentation into multiple distinct ApoBDs[22,29].

In this study, we describe a mechanism of generating large ApoEVs that mark the site of cell death via a cell-retraction dependent process. Upon induction of apoptosis, we showed that adherent cells retract and leave behind actin-rich membrane tracks resembling a cellular 'footprint', coined as the 'FOotprint Of Death' (FOOD), which subsequently round into large ApoEVs (~2 μm in diameter) denoted as FOOD-derived ApoEVs (F-ApoEVs). F-ApoEVs expose the 'eat-me' signal phosphatidylserine (PtdSer) and are cleared from the site of cell death by neighbouring phagocytes. In an infection setting, F-ApoEVs generated from influenza A virus (IAV)-infected apoptotic cells can aid viral propagation to neighbouring cells. Together, this study has identified an alternative mechanism of generating large EVs during apoptosis, which may have implications in efferocytosis and intercellular communication.

## Results

### Apoptotic cells generate a membranous footprint during cell retraction that aids the formation of large ApoEVs

When performing time-lapse confocal microscopy on apoptotic cells, we captured a distinct morphological process. Human A431 squamous epithelial cells treated with a BH3-mimetic cocktail (ABT-737[30,31] and S63845[32,33]) to induce apoptosis exhibited characteristic apoptotic morphologies including cell rounding, membrane blebbing, and PtdSer exposure as indicated by annexin A5 staining (A5) (Fig. 1a). 3D confocal laser scanning microscopy (CLSM) revealed the formation of PtdSer-exposing membranous remnants that appeared during cell retraction, analogous to a 'footprint' of the cell marking the site of cell death (Fig. 1a, b; Supplementary Fig. 1a, b, Supplementary Video 1). This 'footprint' was only observed at the focal plane close to the cover glass (base) and did not extend beyond the original boundary of the cell, distinct from apoptotic protrusions (i.e. apoptopodia) that actively radiate from apoptotic cells[22]. Owing to this, we coined this phenomenon the 'FOotprint Of Death' or 'FOOD'. The formation of FOOD was consistently observed in a variety of cell types including primary human umbilical vein endothelial cells (HUVECs), mouse embryonic fibroblasts (MEFs), and human cervical adenocarcinoma (HeLa) cells (Fig. 1c–e), with the majority (~80-99%) of apoptotic cells forming FOOD (Fig. 1f). Moreover, visualisation of FOOD by scanning electron microscopy (SEM), showed the ultrastructure as thin membrane surrounding the main cell body (Fig. 1c–e). In addition to the

BH3-mimetic cocktail to specifically target the intrinsic apoptotic pathway[30,32], FOOD was generated in response to several apoptotic stimuli including UV irradiation[22,34], DNA damage inducing agents like etoposide[35] and infection with IAV[36] (Supplementary Figs. 2a–c, 3). Apoptosis induction under these conditions was validated by flow cytometry and immunoblot analysis of caspase cleaved proteins such as caspase 3 and Pannexin 1 (PANX1) membrane channels[37,38] (Supplementary Fig. 3). Notably, MEFs that lack the pro-apoptotic proteins Bax and Bak (Bax[-/-]Bak[-/-]) did not generate FOOD following BH3-mimetic cocktail treatment as compared to WT MEFs (Supplementary Fig. 4), further indicating that FOOD is formed specifically during the progression of apoptosis. Extensive quantification of CLSM revealed the median number of 'branches' of membranous material adhered to the substrate, generated by MEF-derived FOOD, to be ~145, with the median branch thickness of ~1.5 μm, occupying an area of ~193.7 μm$^2$ (Fig. 1g), suggesting that MEFs frequently form FOOD composed of many thin membrane structures.

Extracellular matrix (ECM) protein coatings can facilitate cell attachment in vitro and mimic the in vivo cellular environment[39]. Therefore, we examined FOOD on surfaces coated with bovine neutralised type I collagen, human fibronectin, or fibronectin-enriched collagen. MEFs seeded on ECM proteins readily formed FOOD upon the induction of apoptosis (Fig. 1h). Similar findings were also observed for MEFs undergoing apoptosis within a 3D extracellular matrix (Fig. 1i), demonstrating that the formation of FOOD was not restricted to artificial surfaces such as plastic and glass cover slides. Taken together, these findings demonstrate that FOOD formation is a common cellular phenomenon that occurs during apoptosis, and that FOOD can form on surfaces that mimic physiological settings.

To gain a high-resolution temporal insight into FOOD formation, we performed lattice light sheet microscopy (LLSM) on MEFs undergoing apoptosis. LLSM imaging revealed that FOOD is first composed of several flat, sheet-like structures of membrane that gradually 'round up' into discrete vesicle-like structures following exposure of PtdSer on FOOD membranes (Fig. 2a, b, Supplementary Video 2). These vesicles, herein referred to as FOOD-derived ApoEVs (F-ApoEVs), were ~2 μm in diameter (Fig. 2c) with the median number of F-ApoEVs generated per cell within a period of 4 h following apoptosis induction being ~40 (Fig. 2d; Supplementary Fig. 5). Notably, F-ApoEV rounding also readily occurred following apoptosis induction of MEFs seeded on ECM proteins (Supplementary Fig. 6). Furthermore, F-ApoEVs generated from free-GFP expressing cells also harbour GFP in FOOD and F-ApoEVs (Fig. 2e), indicating that the membrane integrity of FOOD and F-ApoEVs is intact as membrane lysis would result in the release of free GFP. Together, this data demonstrates that the formation of FOOD provides a highly effective method of generating large, ApoEVs.

It should be noted that the biogenesis of F-ApoEVs is distinct from the formation of other large EV subsets such as ApoBDs and migrasomes (large EVs generated from trailing edge of cells during migration[40]). First, as F-ApoEVs are generated from the vesicularisation of membranous remnants attached to extracellular substrates following apoptotic cell retraction (Fig. 2a), the biogenesis mechanism is clearly distinct from ApoBDs which are generated from membrane blebs and dynamic apoptopodia that actively radiate from the apoptotic cell[23]. Second, cell types like MEFs do not readily form ApoBDs due to the lack of apoptopodia formation[34], further highlighting that the F-ApoEVs observed are not simply deposition of ApoBDs onto extracellular substrates. Third, cell migration and the presence of membranous remnants prior to apoptosis induction were not observed during time-lapse imaging (Fig. 2f, Supplementary Fig. 7), ruling out the presence of migrasomes. Fourth, FOOD and F-ApoEV formation readily occurred in the presence of both the cell migration inhibitor jasplakinolide[41] (Fig. 2g; Supplementary Fig. 8, 9), and migrasome inhibitors SMS2-IN-1[42] (sphingomyelin synthase 2 inhibitor) and ISA-2011B[43] (PIP5K1α inhibitor) (Supplementary Fig. 10).

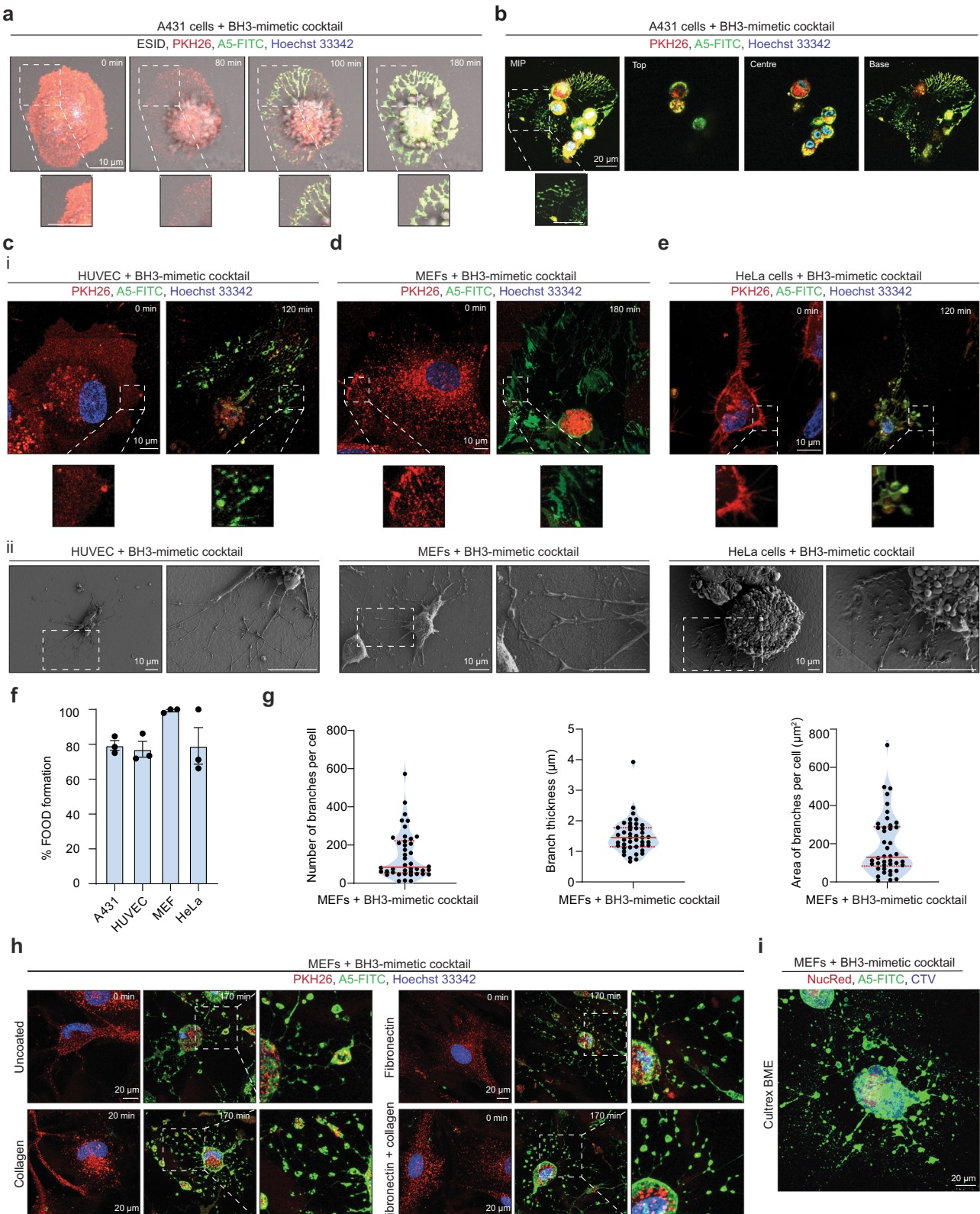

Together, these data support the concept of FOOD formation and subsequent vesicularisation as an alternative mechanism for generating large ApoEVs during apoptosis (Fig. 2h).

## FOOD/F-ApoEVs are enriched in actin and adhesion proteins

Identification of EV contents can infer mechanistic and functional insights. Therefore, we next performed label-free quantitative proteomic analysis on A431-derived FOOD. During our investigation, we noted that FOOD/F-ApoEVs are not readily displaced from the substrate (i.e. plastic cell culture dish) by trypsin/EDTA treatment. Based on this feature, we developed an in situ approach to enrich FOOD on the culture dish by removing apoptotic cells and other EV subsets in the culture supernatants with a series of PBS wash steps, before removing remaining cells with trypsin/EDTA treatment

**Fig. 1 | Apoptotic adherent cells generate a membranous 'footprint of death' during cell retraction. a**, **b** A431 epithelial cells were treated with a BH3-mimetic cocktail (2.5 μM ABT-737, 0.5 μM S63845) and imaged by time-lapse confocal laser scanning microscopy (CLSM). Cell membrane and nucleus were visualised by PHK26 (red) and Hoechst 33342 (blue) staining, respectively, and exposed phosphatidylserine (PtdSer) was examined using A5-FITC (green). Cells in (**b**) are representative of cells 180 min post apoptosis induction. Maximum intensity projection (MIP) shown (left) and Z-stack images of top, centre and base of cells shown (left to right). Images demonstrating morphological changes following apoptosis induction in (**c**) HUVEC (2.5 μM ABT-737, 0.5 μM S63845), **d** MEF (5 μM ABT-737, 10 μM S63845), and **e** HeLa cells (2.5 μM ABT-737, 0.5 μM S63845) following treatment with a BH3-mimetic cocktail. Representative CLSM i (n = 3) and scanning electron microscopy (SEM) ii (n = 2). **f** Quantification of the percentage of apoptotic A431, HUVEC, MEF, and HeLa cells that form FOOD during the progression of apoptosis. Data is pooled from (n = 3) independent experiments. Apoptotic cells were identified based on A5 staining and apoptotic morphologies (i.e. cell retraction and membrane blebbing). Error bars represent s.e.m. **g** Quantitative analysis of the FOotprint Of Death (FOOD) generated by MEFs: number of branches per cell, branch thickness (μm), and area of branches per cell (μm²). Data is pooled from (n = 3) independent experiments. **h** Representative time-lapse CLSM MIP of MEFs on uncoated, collagen, fibronectin, and fibronectin and collagen coated chamber slides, treated with a BH3-mimetic cocktail. **i** Representative CLSM MIP of MEF in a Cultrex BME 3D-matrix. Cell cytosol and nucleus were visualised by cell trace violet (blue) and NucRed staining, respectively, and PtdSer exposed was examined using A5-FITC. At least three independent experiments were performed for all experiments unless otherwise specified.

(Supplementary Fig. 11a–c). The FOOD on the culture dish was then directly lysed for proteomic analysis. We identified a total of 601 proteins present in FOOD, 573 of which were also found in apoptotic cells and 28 proteins unique to FOOD (Fig. 3a, Supplementary File). The most abundant FOOD proteins were actin and histone proteins, as well as several cytoskeletal, membrane, and adhesion proteins. Gene ontology analysis revealed an abundance of proteins found in FOOD that can bind to integrins, cadherin, and actin, as well as protein components of the cytoskeleton and focal adhesions (Fig. 3b, Supplementary Fig. 12a).

In support of the proteomic data, we performed CLSM analysis and confirmed the presence of F-actin in FOOD and F-ApoEVs by SiR-actin and phalloidin staining (Fig. 3c, d). Time-lapse imaging revealed that when the apoptotic cell contracts to generate FOOD, F-actin can be found within F-ApoEVs during vesicle rounding (Fig. 3e). To further visualise the localisation of F-actin in FOOD relative to the morphological features observed in SEM, correlative light and electron microscopy (CLEM) was performed and demonstrated that the thin-fibrous structures surrounding apoptotic cell to harbour F-actin (Figs. 1c, 3f). In addition to F-actin, we also observed the presence of other cytoskeletal components such as tubulin in FOOD and F-ApoEVs (Fig. 3g). Gene ontology analysis indicated a large proportion of proteins enriched in FOOD are components of focal adhesions such as filamin, α-actinin, various integrins, and vinculin (Supplementary Fig. 12b). The presence of vinculin in FOOD and F-ApoEVs was confirmed by immunostaining, however, the localisation of vinculin is more disperse compared the discrete puncta observed in viable cells (Fig. 3h), possibly due to caspase-mediated cleavage of focal adhesion complexes[44].

Large EV subsets such as ApoBDs, migrasomes and exophers often contain organelles such as mitochondria, nucleus and golgi[7,13,45,46]. Utilising cell lines expressing fluorescently-tagged proteins to monitor the localisation of golgi and lysosomes, and staining cells with MitoTracker Green to track mitochondria, we found that organelles are present in low abundance, with on average, less than 20% of FOOD harbouring organelles (Supplementary Fig. 13a–g). Although histones were identified in our proteomic analysis of FOOD (Fig. 3b), we were unable to detect histones in FOOD by microscopy approaches (Fig. 3d, Supplementary Fig. 13h). Furthermore, nuclear DNA was not present in FOOD as monitored by Hoechst 33342 staining (Supplementary Fig. 1b). This data indicates FOOD is devoid of nuclear materials and the that the enrichment of histones in the proteomic analysis may be due the release of histones during apoptosis[47] and subsequent deposition onto the culture dish.

### Formation of F-ApoEVs through FOOD is dependent on ROCK1 activation

During apoptosis, caspase-activated ROCK1 aids both cell contraction[26] and plasma membrane blebbing[24]. Since cell contraction is involved in FOOD formation (Fig. 1a), we examined the role of ROCK1 in this process. To this end, we utilised MEFs derived from ROCK1 non-cleavable (ROCK1nc) mice that express a mutated form of ROCK1 resistant to caspase-mediated cleavage, which results in reduced regulatory myosin light chain phosphorylation and consequent decreased cellular contractile force generation during apoptotic cell death[48] (Supplementary Fig. 14). Upon apoptosis induction, ROCK1nc MEFs exhibited defective FOOD formation, whereby a greater proportion of the cell body remained adhered to the surface of the substrate without cell retraction (Fig. 4a). ROCK1nc MEF-derived FOOD exhibited flat, membrane-sheet like morphology (Fig. 4a). Furthermore, quantification of F-ApoEV formation revealed that ROCK1nc MEF-derived FOOD rounds into significantly fewer F-ApoEVs, compared to the WT control (Fig. 4b). Further quantification of CLSM analysis confirmed that ROCK1nc cells have a significantly greater number of FOOD branches compared to WT cells and the branches occupied a larger area (Fig. 4c), highlighting a defect in cell retraction for ROCK1nc cells undergoing apoptosis. Defects in FOOD morphology in ROCK1nc MEFs were also confirmed by SEM analysis (Fig. 4d). Time-lapse CLSM analysis of F-actin in apoptotic ROCK1nc MEFs revealed a greater amount of F-actin in FOOD generated by ROCK1 MEFs compared to WT cells (Fig. 4e), potentially limiting vesicularisation into abundant F-ApoEVs. Together, this data reveals that caspase-mediated activation of ROCK1 is required for FOOD formation, and abundant F-ApoEV generation.

### F-ApoEVs can be efferocytosed and traffic viral material

Lastly, we explored the functional consequence of FOOD formation. As FOOD exposes the 'eat-me' signal PtdSer (Fig. 1a), we examined whether FOOD could be recognised and efferocytosed by phagocytes using LLSM. In co-culture studies of bone marrow-derived macrophages (BMDMs) and isolated MEF-derived FOOD, BMDMs could interact with FOOD and engulf F-ApoEVs (Fig. 5a, Supplementary Fig. 15). Once at the site of cell death, BMDMs could be observed 'grabbing' strands of FOOD and efferocytosing F-ApoEVs. 3D analysis of LLSM data clearly demonstrated the internalisation of A5⁺ F-ApoEVs inside BMDMs, indicative of efferocytosis (Supplementary Video 3, 4). As FOOD resembles a smaller 'bite-sized' portion of the apoptotic cell, we hypothesised that the FOOD may serve as an 'appetizer' to prime phagocytes before engulfing the whole apoptotic cell. Therefore, we primed BMDMs with MEF-derived FOOD for 24 h and subsequently performed an efferocytosis assay with CypHer 5E-labelled apoptotic Jurkat T cells. In contrast to the untreated BMDMs, BMDMs primed with FOOD exhibited an increase in efferocytotic efficiency (Fig. 5b). Taken together, these results indicate that FOOD may aid phagocytes to identify the site of cell death and prepare for efferocytosis.

IAV infection induces vast amounts of apoptosis in vivo, which can result in the displacement of cells from the respiratory track and lung[36]. Therefore, IAV infection may resemble a physiologically relevant model where apoptotic cells could leave behind FOOD at the site of cell death. Thus, we next investigated the functional consequence of FOOD formation under a pathological context using the PR8 H1N1 model of IAV infection. A549 epithelial cells readily generated FOOD in response to IAV-induced apoptosis whereby FOOD exhibited PtdSer-rich membrane depositions with thin actin fibres and resulted in

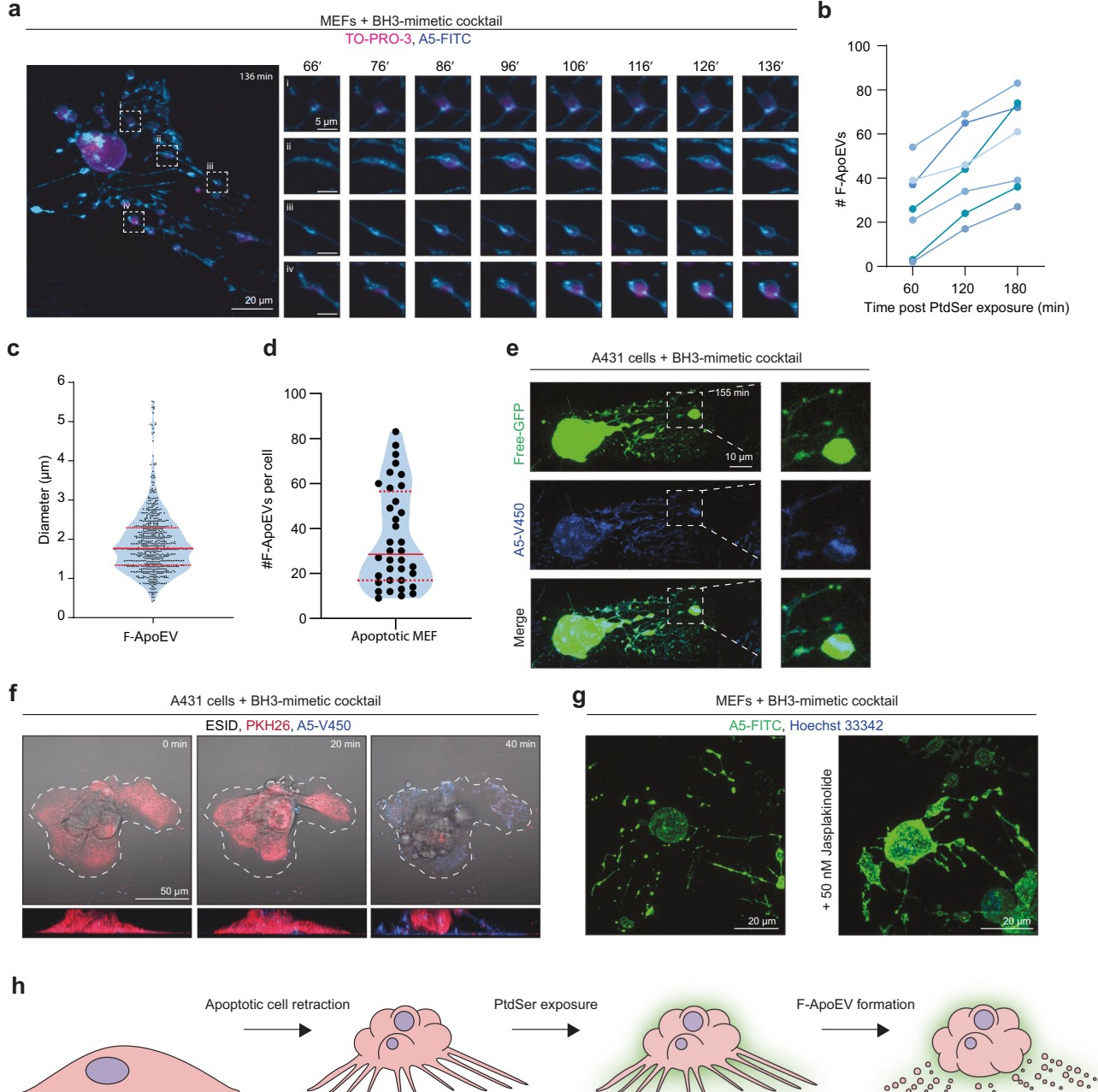

**Fig. 2 | Formation of large ApoEVs from FOOD. a** Representative maximum intensity projections (MIP) images from time-lapse lattice light sheet microscopy (LLSM) of FOOD-derived ApoEVs (F-ApoEV) formation in MEFs following treatment with a BH3-mimetic cocktail (5 μM ABT-737, 10 μM S63845). Cell shown left is representative of 136 min post apoptosis induction. Rounding of deposited membrane into individual F-ApoEVs shown right, 10 min intervals from 66 min to 136 min post apoptosis induction. Cells were stained with A5-FITC (blue) and TO-PRO-3 (magenta). Data is representative of (*n* = 3) independent experiments. **b** Quantification of the rate of vesicle formation from LLSM images at 60-, 120-, and 180-min post initial phosphatidylserine (PtdSer) exposure in FOOD. Data points represent individual cells (*n* = 7), representative from (*n* = 3) independent experiments. **c** Quantification of F-ApoEVs diameter (μm) from LLSM imaging, 240 min post apoptosis induction. Data points represent individual F-ApoEVs (*n* = 659 F-ApoEVs, from *n* = 18 cells) representative from (*n* = 3) independent experiments. Solid red line indicated mean, dashed red line indicates quarterlies.

**d** Quantification of the number of F-ApoEVs generated per cell from LLSM imaging, 240 min post apoptosis induction. Data points represent individual cells (*n* = 31). Solid red line indicated mean, dashed red line indicates quarterlies. **e** Representative MIP confocal laser scanning microscopy (CLSM) images of F-ApoEV formation in free GFP expressing A431 cells 155 min post-treatment with BH3-mimetic cocktail (2.5 μM ABT-737, 0.5 μM S63845). A5-V450 staining shown in blue. **f** Representative time lapse CLSM images of A431 cells following treatment with a BH3-mimetic cocktail. Lower Z-plane shown above, and 3D rendered cross-section of the orthogonal planes of side view shown below. Cell membranes were visualised with PHK26, and exposed PtdSer with A5-V450. **g** Representative CLSM MIP of apoptotic MEFs treated with Jasplakinolide (50 nM) or vehicle control. Exposed PtdSer was visualised with A5-FITC and cell nucleus with Hoechst 33342. **h** Schematic diagram of F-ApoEV formation from FOOD. At least three independent experiments were performed for all experiments unless otherwise specified.

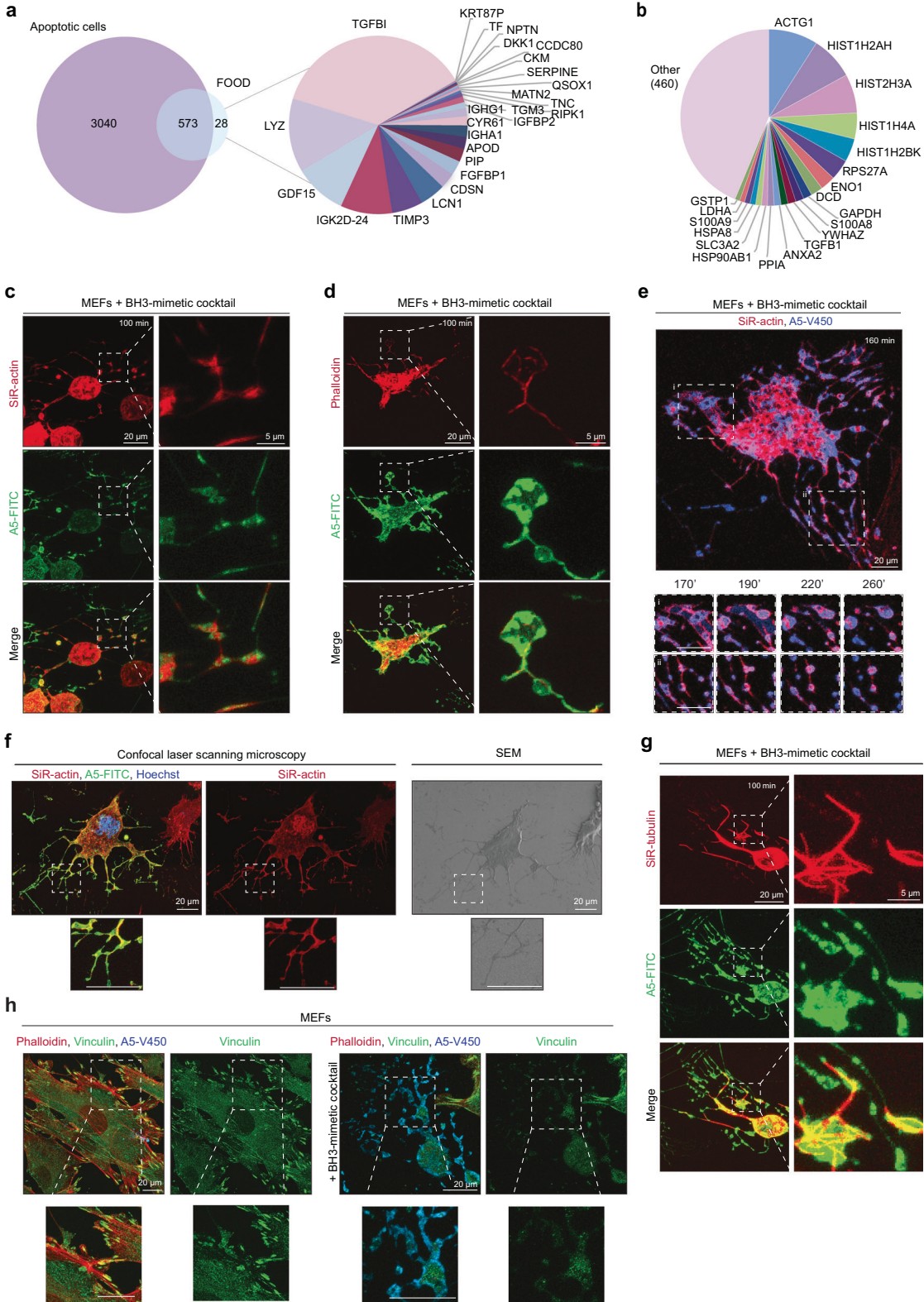

F-ApoEV formation (Fig. 5c). Similar findings were also observed in primary murine BMDMs infected with IAV (Supplementary Fig. 16). SEM analysis revealed that the ultrastructure of FOOD in response to IAV is comparable to FOOD formed under other apoptotic settings (Fig. 5d). Furthermore, viral proteins such as nucleoprotein (NP) and hemagglutinin (HA) were detected within A549-derived FOOD and F-ApoEVs, 24 h post infection (Fig. 5e, f). Importantly, TEM demonstrated that on occasion, IAV viral particles could be observed in

F-ApoEVs isolated from IAV-infected A549 cells (Fig. 5g). To determine if F-ApoEVs generated during IAV infection had the capacity to propagate viral infection, we co-cultured F-ApoEVs isolated from IAV-infected A549 cells with healthy A549 lung epithelial and observed a marked increase in A549 cells staining positive for viral NP, indicative of IAV infection (Fig. 5h, Supplementary Fig. 17). Notably, approximately half of the infected A549 cells were undergoing cell death as indicated by A5 staining (Fig. 5i). Thus, F-ApoEVs derived from IAV-

**Fig. 3 | FOOD is enriched in adhesion associated proteins and cytoskeletal components. a** Venn diagram (left) showing the number of proteins unique to and shared by apoptotic A431 cells and A431 cell-derived Footprint Of Death (FOOD) as identified by proteomic analysis, and pie chart (right) showing the 28 proteins unique to FOOD, organised by relative abundance. **b** Pie chart showing the top 20 most highly abundance proteins present in FOOD as identified via proteomic analysis, organised by relative abundance. **c, d** Representative maximum intensity projection (MIP) confocal laser scanning microscopy (CLSM) images of FOOD formation in MEFs 100 min post-treatment with BH3-mimetic cocktail (5 μM ABT-737, 10 μM S63845). Cell actin was visualised by SiR-Actin (red) (**c**) or Phalloidin (red) (**d**), and exposed phosphatidylserine (PtdSer) visualised with A5-FITC. **e** Representative MIP images from time-lapse CLSM of FOOD-derived ApoEV (F-ApoEV) formation in MEFs following treatment with a BH3-mimetic cocktail.

Rounding of deposited membrane into individual F-ApoEVs at 20 min intervals from 170 to 260 min post apoptosis induction, shown below, as indicated by the dotted box. Cell actin was visualised by SiR-Actin and exposed PtdSer visualised with A5-V450. **f** Representative correlative light and electron microscopy (CLEM) images of FOOD formation in MEFs 120 min post-treatment with BH3-mimetic cocktail. CLSM images shown left, and correlating scanning electron microscopy (SEM) image shown right. **g** Representative MIP CLSM images of FOOD formation in MEFs 100 min post-treatment with BH3-mimetic cocktail. Microtubules were visualised by SiR-Tubulin (red) and exposed PtdSer visualised with A5-FITC. **h** Localisation of vinculin (green) in viable (left) or BH3-mimetic cocktail (2 h post treatment) treated. Cell cytoskeleton visualised with phalloidin and exposed PtdSer with A5-V450. At least three independent experiments were performed for all experiments unless otherwise specified.

infected cells may serve as a reservoir for viral proteins and infectious virions and contribute to viral spread.

## Discussion

Apoptosis occurs throughout life in essentially all tissues as part of normal development and cell turnover. Since >200 billion cells die in the human body via apoptosis daily, it is critical for phagocytes to detect the location of cell death to aid the removal of apoptotic materials effectively. Here, we describe a mechanism by which the dying cells could leave behind a 'FOotprint Of Death' (FOOD) to mark the site of cell death. Importantly, through the generation of FOOD, a distinct subclass of large ApoEVs is formed, coined as FOOD-derived ApoEVs (F-ApoEVs), which can mediate important communication with phagocytes at the site of cell death. The data presented here provide several key insights into the highly orchestrated process of apoptotic cell-phagocyte interaction.

During the progression of apoptosis, a distinct subset of large EVs known as ApoBDs are generated though highly coordinated morphological changes. This process involves the formation of membrane blebs, followed by apoptopodia protruding out from the cell body, and subsequent fragmentation of apoptopodia to generate discrete ApoBDs[7,22,23]. These ApoBDs can often dissipate from the site of cell death and interact with cells within the vicinity[49,50]. In stark contrast to ApoBDs, we found that prior to apoptotic membrane blebbing, adherent cells can retract from the extracellular substrate and generate another distinct subset of large EVs, described here as F-ApoEVs, that are tightly anchored to the site of cell death. The formation of F-ApoEVs also progresses through highly coordinated morphological stages including: 1) Cell retraction: the apoptotic cell retracts and deposits membranous material and long F-actin fibres on the extracellular matrix, 2) PtdSer exposure: the gradual exposure of PtdSer on FOOD membranes, and 3) EV rounding: FOOD membrane undergoes a 'rounding' event to form distinct EVs. Thus, the generation of F-ApoEVs is morphologically and spatially distinct from ApoBD formation. As F-ApoEVs can remain attached to the site of cell death, it may serve a specific function during the clearance of apoptotic cells. Notably, F-ApoEVs can expose the 'eat-me' signal PtdSer, interact with and be engulfed by phagocytes. Furthermore, since F-ApoEVs are largely intact, these apoptotic materials may harbour metabolites and intracellular proteins that could function as 'good-bye' signals and damage-associated molecular patterns, respectively, to initiate downstream processes such as wound healing and inflammation at the site of cell death[1,2,10]. Notably, engulfment of apoptotic cells has been shown to promote continual efferocytosis of macrophages through the processing of apoptotic cell-derived amino acids[51]. Similarly, priming of BMDMs with FOOD promoted efferocytosis, supporting the concept that leaving behind FOOD at the site of cell death can also contribute to the overall clearance process.

The capacity for EVs to exert effects on recipient cells is underpinned by their cargo. It's well established that EV biogenesis pathways influence the packing of specific cargo into EVs. For example, the incorporation of endosome-associated proteins into exosomes is driven by their origins in the endosomal pathway[52,53]. Since F-ApoEVs and ApoBDs are generated via distinct morphological changes during apoptosis, these large ApoEV subsets may package distinct cellular components. Notably, while a number of studies have demonstrated that nuclear contents, mitochondria and Golgi are packaged into ApoBDs[7,13,22], these organellar contents are rarely found in F-ApoEVs. This observation could be due to the majority of cellular contents being retracted with the main cell body during apoptosis, with only limited membrane and cytoskeletal components depositing on the substrate. Nevertheless, in the context of IAV infection, viral proteins and infectious virions were found in FOOD and F-ApoEVs, highlighting the potential role of these apoptotic cell-derived materials in mediating viral propagation as well as antigen presentation. Consistent with these findings, other subtypes of ApoEVs such as ApoBDs has also been shown to harbour infectious IAV[8] and African swine fever virus[15], suggesting the ability of viruses to hijack multiple ApoEV formation pathways to propagate viral infections.

Similar to the formation of F-ApoEVs through FOOD, several studies have demonstrated that healthy cells can also transfer adherent membranous material through cell retraction, including the deposition of retraction fibres on the substrate during migracytosis[40], during the formation of neutrophil trails[54], and the generation of T cell microvilli particles (TMPs) on neighbouring cells[55]. In these processes, membrane deposited on a substrate during migration or following cell-cell contact round into adherent EVs, which remain attached to site of generation. Notably, biogenesis of these contact-dependant structures is underpinned by cell-adhesion[40,54,55], and for migrasomes, is also regulated by ROCK1[56]. While the detachment of the apoptotic cell from the extracellular substrate is thought to be due to caspase-mediate cleavage of focal adhesion sites[57–59] and cleavage of ROCK1 driving actomyosin contraction, the precise mechanism by which cells detach from the substrate during apoptosis remains unclear[60]. Moreover, the molecular fate of focal adhesion complex protein components following their breakdown, i.e., adhered to the extracellular substrate or the main cell body, is unknown. Here, we demonstrate FOOD formation is dependent on ROCK1 activation during apoptosis, and FOOD contains actin and tubulin, as well as several focal adhesion proteins including vinculin, talin, and integrins. This indicates that following ROCK1-mediated apoptotic cell retraction, components of the focal adhesion complexes may remain at the site of cell death localised with FOOD. However, whether these proteins are functional and contribute to the adherence of F-ApoEVs to the extracellular substrate remains to be defined. It is important to note that although the formation of FOOD and F-ApoEVs exhibit morphologic and mechanistic (e.g. ROCK1-regulated) similarities to the generation of migrasomes during cell migration[56], as well as non-canonical migracytosis (i.e. formation of migrasome in viable cells in the absence of cell migration such as after exposure to the bacterial toxin TcdB3 from *C. difficile*[61]), FOOD/F-ApoEVs were observed specifically during the progression of apoptotic

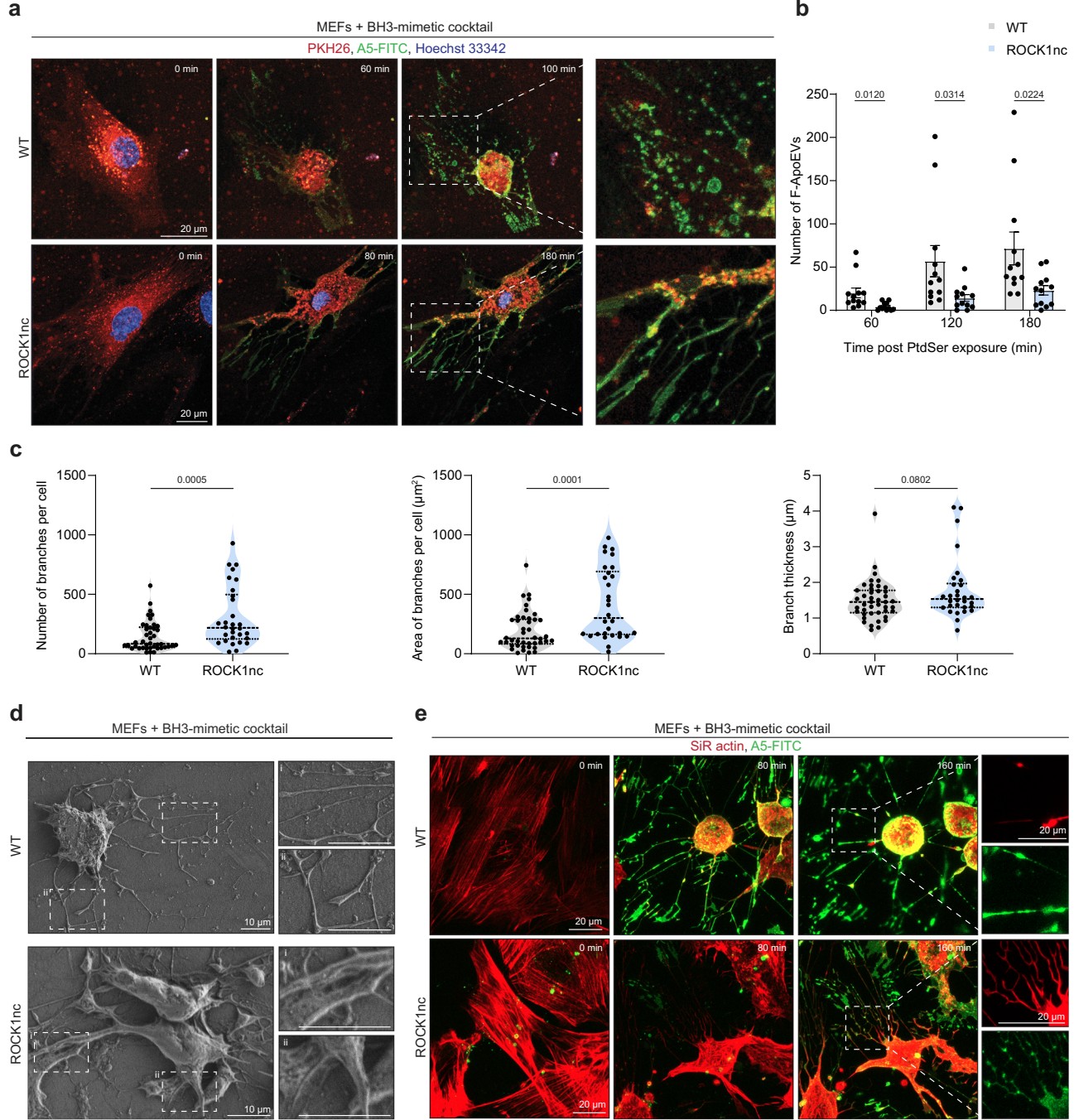

**Fig. 4 | Formation of FOOD and F-ApoEVs is dependent on caspase-cleaved ROCK1. a** WT or ROCK1nc MEFs were treated with a BH3-mimetic cocktail (5 μM ABT-737, 10 μM S63845) and imaged by time-lapse confocal laser scanning microscopy (CLSM). Cell membrane and nucleus were visualised by PHK26 and Hoechst 33342 staining, respectively, and exposed phosphatidylserine (PtdSer) with A5-FITC. **b** Quantification of the number of FOOD-derived ApoEV (F-ApoEVs) generated per cell from CLSM imaging, by WT (grey) or ROCK1nc (blue) MEFs, 60-, 120-, and 180 min post initial PtdSer exposure. Data points represent individual cells ($n = 12$). **c** Quantitative analysis of The FOotprint Of Death (FOOD) generated by WT or ROCK1nc MEFs: number of branches per cell, area of branches per cell ($\mu m^2$), and branch thickness ($\mu m$). Data is pooled from ($n = 3$) independent experiments. **d** Representative scanning electron microscopy (SEM) images of WT or ROCK1nc MEFs, treated with a BH3-mimetic cocktail. ($n = 2$). **e** Visualisation of actin in FOOD/F-ApoEV formation in WT or ROCK1nc MEFs following treatment with a BH3-mimetic cocktail. Representative maximum intensity projection (MIP) images from time-lapse CLSM shown. Cell actin was visualised by SiR-Actin and exposed PtdSer with A5-FITC. Error bars in (**b**, **c**) represent s.e.m. Unpaired student's two tailed *t*-test was performed to determine the indicated *p*-values.

cell death in the absence of cell migration, highlighting the biogenesis of migrasomes and FOOD/F-ApoEVs are distinct. Furthermore, it is worth noting that migrasome-specific protein markers identified via quantitative mass spectrometry (i.e. NDST1, PIGK, CPQ, and EOGT[62]) were not detected in proteomic analysis conducted on FOOD/F-ApoEVs described herein. Collectively, the ability of healthy and dying cells to leave behind membranous packages at sites where the cells once travelled or occupied highlights the important of these processes in intercellular communication.

## Methods

### Ethics approval

All animal studies were approved by the La Trobe University Animal Ethics Committee or the Water and Eliza Hall animal ethics committee

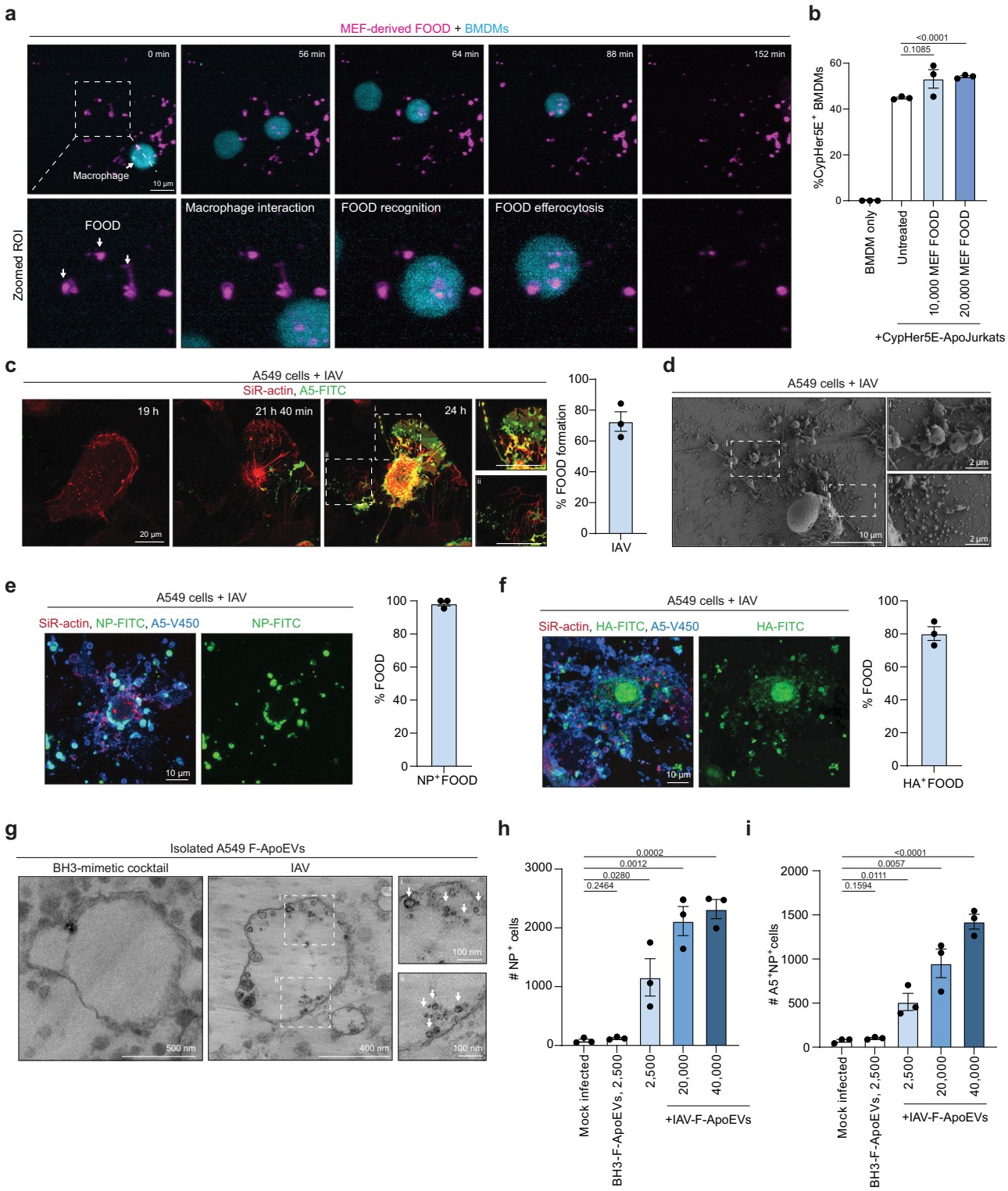

and performed in accordance with the Australian code for the care and use of animals for scientific purposes.

## Reagents

Donkey Anti-Mouse IgG Alexa Fluor-488, A5-FITC, A5-V450, A5-PE and A5 binding buffer were purchased from BD Bioscience. TO-PRO-3, LIVE/DEAD APC, CellTrace Violet, MitoTracker Green, Phalloidin 647, NucRed Live 647, Hoechst 33342, and Collagenase type 1 were purchased from ThermoFisher Scientific. Jasplakinolide, Anti-Vinculin antibody, PKH26, and PKH67 were purchased from Sigma-Aldrich. Q-VD-OPh was purchased from Merk. SiR-Actin and SiR-Tubulin were purchased from Spirochrome. ABT-737, S63845, SMS2-IN-1 and ISA-2011B were purchased from MedChem Express. Etoposide was purchased from Selleckchem. Type 1 Collagen Solution, PureCol® EZ Gel, and Fibronectin were purchased from Advanced Biomatrix. Cultrex® Basement Membrane Extract was purchased from R&D Systems. AccuCount Blank particles was purchased from Spherotech. Procure 812 resin kit, TEM grids, osmium tetroxide, uranyl acetate and lead citrate were purchased from ProSciTech.

**Fig. 5 | The role of F-ApoEVs in cell clearance and intercellular communication.**
**a** Representative maximum intensity projection (MIP) images from time-lapse lattice light sheet microscopy (LLSM) of cell trace violet-stained BMDM interacting with and engulfing MEF-derived FOOD/F-ApoEVs stained with A5-PE (magenta).
**b** Quantification of re-feeding engulfment assay measured by %Cypher5E$^+$ BMDMs following initial incubation with the FOotprint Of Death (FOOD)/FOOD-derived ApoEVs (F-ApoEVs) derived from $1 \times 10^4$ or $2 \times 10^4$ MEFs for ~24 h, followed by incubation of CypHer5E-labelled apoptotic Jurkat T cells. Data is representative of ($n = 3$) independent experiments. **c** Representative time-lapse MIP confocal laser scanning microscopy (CLSM) images of FOOD formation in A549 cells 19- to 24 h post-infection (p.i.) with influenza A virus (IAV) (MOI = 10) (right). Quantification of the frequency of FOOD formation in IAV-infected apoptotic cells (left). Data is representative of ($n = 3$) independent experiments. **d** Representative scanning electron microscopy (SEM) images of FOOD formation in IAV-infected apoptotic cells (24 h p.i.). Data is representative of ($n = 2$) independent experiments. Localisation of viral nucleoprotein (NP) (**e**) or hemagglutinin (HA) (**f**) (green) in IAV-infected A549 cell-derived FOOD/ F-ApoEVs (24 h p.i.). Quantification of %FOOD stained positive for NP or HA protein shown on the right. **g** Representative transmission electron microscopy (TEM) images of isolated F-ApoEVs from A549 cells induced to undergo apoptosis with either BH3-mimetic cocktail treatment, or IAV infection. White arrows indicate virions. Quantification of the absolute number of NP$^+$ A549 cells (**h**) or NP$^+$A5$^+$ A549 cells (**i**) following co-incubation of A549 cells with the indicated amount of F-ApoEVs isolated from BH3-mimetic cocktail treated or IAV infected A549 cells. Data is representative of ($n = 3$) independent experiments. Error bars represent s.e.m. Unpaired student's two tailed $t$-test was performed to determine the indicated $p$-values.

## Cells

Human A431 cells were cultured in MEM with L-glutamine, HeLa, Jurkat T cells and A549 alveolar basal epithelial cells were cultured RPMI 1640, MEFs were cultured in DMEM (Gibco). Media was supplemented with 10% (v/v) foetal calf serum, 50 IU/mL penicillin and 50 μg/mL streptomycin, and 0.2% (v/v) MycoZap. Human umbilical vein endothelial cells (HUVECs) were cultured in Endothelial Cell Basal Medium-2, supplemented with EHM-2 Bullet Kit (Lonza). ROCK1wt and ROCK1nc MEFs were generated from 14.5-day old embryos. Bax$^{-/-}$Bak$^{-/-}$ MEFs were a kind gift from Prof David Huang (The Walter and Eliza Hall Institute of Medial Research). BMDMs were differentiated using GM-CSF (Miltenyi Biotec) and used between day 10 − 12 of differentiation, or with L929 media (containing M-CSF) and used between day 6-7. All cell lines were incubated at 37 °C with humidified 5% CO$_2$.

## Induction of apoptosis

To induce apoptosis by UV irradiation, MEFs were exposed to 150 mJ/cm$^2$ of irradiation using the Stratagene UV Stratalinker 1800 (Agilent Technologies) and incubated at 37 °C with humidified 5% CO$_2$. For drug induced apoptosis, cells were treated with BH3-mimetics ABT-737 and S63845 (BH3-mimetic cocktail, small molecule inhibitors that specifically targets the pro-survival proteins Bcl-2 and Bcl-xL, and MCL-1, respectively[30–33]) at indicated concentrations, or etoposide (topoisomerase II inhibitor[35], 250 μM). Where specified, cells were additionally treated with migration inhibitor jasplakinolide (50 nM)[41], migrasome inhibitors SMS2-IN-1 (30 μM) or ISA-2011B (20 μM), or pan-caspase inhibitor Q-VD-OPh (50 μM) during apoptosis.

## IAV infection

WT-PR8 (A/Puerto Rico/8/1934 H1N1) (a kind gift from Professor Wei-san Chen) was propagated in SPF embryonated hen eggs. Viral titres were determined as previously described[8]. A549 epithelial cells were infected at a multiplicity of infection of 10 in acidified media for 1 h and incubated for 24 h in 5% RPMI unless otherwise specified.

## Confocal microscopy

Cells were prepared for microscopy by seeding in 8-well Nunc® Lab-Tek® II chamber slides (Nunc, Denmark) in complete media overnight (37 °C, 5% CO$_2$), prior to staining and treatment. Where specified, the plasma membrane was stained with PKH26 or PKH67 (1:200 for 5 min), and nucleus with Hoechst 33342 (1 μg/mL) or NucRed 647, according to the manufacturers protocol. To monitor PtdSer exposure cells were stained with A5-FITC, A5-V450 or A5-PE (1:100). Where specified, cells were stained using CellTrace Violet according to the manufacturers protocol. To examine the localisation of the cytoskeleton, cells were seeded onto collagen coated slides then stained with SiR-actin or SiR-tubulin (1:1000; Spirochrome, 4 h, 37 °C, 5% CO$_2$). To examine the localisation of viral proteins, IAV-infected A549 cells were stained with NP-FITC (1:50; Invitrogen, D67J, MA1-7322 or Saphire Bioscience, 1331, GTX36902), or HA-FITC (1:200; Santa Cruz Biotechnology, 12CA5, sc-57592) ~24 h p.i. To examine the localisation of vinculin in FOOD, cells were probed with primary anti-vinculin antibody (1:200; hVIN-1 Sigma-Aldrich, V9264) and secondary donkey anti-mouse IgG H&L Alexa Fluor-488 (1:400; Abcam, ab150105). Where specified, A431 cells were transfected with free-GFP, clover-sialyltransferase (clover-SiT), or clover-lysosomal associated membrane glycoprotein 1 (clover-LAMP1) and imaged as described above. Confocal microscopy was performed using the Zeiss 800/900 Confocal Laser Scanning Microscope with Zen (v 3.10) software (Carl Zeiss, Germany) using a 63×1.4 NA magnification lens, and for live imaging, with oil at 37 °C and 5% CO$_2$. Image processing and analysis were performed Zen imaging software Zen (v 3.10) or Fiji (v 2.2.0).

## Lattice light sheet microscopy

To capture FOOD formation by LLSM, 5000 MEFs were seeded into wells of an ibidi μ-Slide 8 Well Glass Bottom chamber slide. The next day, MEFs were washed with PBS and stained with PKH26 (1:200 for 5 min). Remaining dye was quenched with BSA and cells were washed. A stain and BH3-mimetic cocktail mix was then added directly to the MEFs including AV-FITC (1:100), TO-PRO-3 (1:1000), S63845 (10 μM) and ABT-737 (5 μM). Time-lapse live-cell data were acquired using a Lattice Light Sheet 7 (Zeiss). Light sheets (488 nm, 561 nm and 640 nm) of 30 μm in length with a thickness of 1 μm were created at the sample plane via a 13.3× 0.44 numerical aperture (NA) objective. Fluorescence emission was collected via a 44.83 × 1 NA detection objective via a multiband stop, LBF 405/488/561/633, filter. Aberration correction was set to a value of 170 to minimize aberrations as determined by imaging the Point Spread Function using 100 nm fluorescent microspheres. Resolution was determined to be 454 nm (lateral) and 782 nm (axial). Data were collected with a frame time of 5 ms and a z step of 0.4 μm with 1751 frames per 750 μm by 290 μm. Data were subsequently deskewed and then deconvolved using a constrained iterative algorithm with 15 iterations. Tile regions were imaged for 4−6 h with 2 min intervals at 37 °C, 5% CO$_2$.

## Quantification of FOOD

Fiji (v 2.2.0) was used for FOOD quantification. Briefly, a maximum intensity projection was created from the 3D CLSM image. The annexin channel was used to train a pixel classifier in ilastik[63] to create a cell mask. This cell mask was used for further analysis in Fiji. To keep only the soma and remove the small objects, a binary close operation, followed by a morphological opening filter (radius = 7, element = Disk[64]) was applied to the cell mask. Any remaining branches or large objects were removed manually in Fiji to create a binary image containing only cell soma and no branches. The DAPI channel was binarized using an "Automatic Threshold" and each cell soma was verified to contain DAPI using the labelOverlapCountMap function in CLIJ2[65]. The cell mask from ilastik was subtracted from the binary cell soma mask to keep FOOD associated with each cell. In Fiji, for each cell, the FOOD image was skeletonized and the branching information analyzed using the Skeletonize and Analyze Skeleton function, respectively[66]. The thickness of the branches was measured using the local thickness plugin[67],

and the mean intensity value extracted for each cell as the mean branch thickness. This processing was performed for the ROCK1wt and ROCK1nc MEF CLSM data (n = 3).

## Scanning electron microscopy

Cells were seeded onto ACLAR film pre-coated with bovine Neutralised Type I Collagen Solution (0.5 mg/mL) prior to apoptosis induction. Cells were then fixed with 2.5% Glutaraldehyde in 0.1 M PBS buffer for 1 h at RT, and incubated overnight, 4 °C in PBS. Cells were post-fixed in 1% Osmium tetroxide/ PBS for 1 h at RT, then washed distilled water (2 ×10 min). Next, cells were dehydrated in an ethanol gradient (20%, 50%, 70%, 90%) for 10 min followed by 100% ethanol (3 ×10 min). Samples were dried with hexamethyldisilazane and imaged on a Hitachi SU7000 FE SEM running at 1 kV.

## Correlative light and electron microscopy

FlixiPERM® re-usable silicone 8-well inserts were attached to glass coverslips, prior to coating with bovine Neutralised Type I Collagen Solution (0.5 mg/mL) and cells seeding. MEFs were stained with SiR-Actin, A5-FITC, and Hoechst 33342 as above, treated with a BH3-mimetic cocktail and following 120 min incubation directly fixed in the chamber with 4% paraformaldehyde for 20 min to preserve cellular structures. Following fixation, cells were imaged using confocal microscopy as above for approximately 60 min. As the sample was fixed prior to confocal imaging, CLSM and SEM images are considered to be taken at the same timepoint. To allow for confocal images to be correlated during SEM, a 41 × 5 tile region image of the entire width of the top of the well (9.46 mm × 1.18 mm) was obtained during confocal microscopy using the 20 × 0.8 NA objective. Immediately following confocal imaging, the FlixiPERM® re-usable silicone insert was removed from the glass coverslip, the orientation marked, and the sample prepared for SEM and imaged as above. To correlate the CLSM during SEM imaging, the CLSM 41 ×5 tile region image was imported into the SEM software as a navigation image and aligned with the SEM sample by manually identifying and subsequently selecting 3 morphologically distinct cells in the sample with corresponding cells in the CLSM tile region. Of note, due to repeated buffer exchanges during the SEM processing procedure, non-adherent or weakly-adherent structures present in the CLSM image may have detached prior to SEM image. Image process and analysis was performed using ZEN imaging software, and correlation performed using Fiji (v 2.2.0).

## ECM surface coating

ECM surface proteins were coated on glass-bottom 8-well chamber slides (Nunc, Denmark). For collagen coating, slides were treated with bovine Neutralised Type I Collagen Solution (100 μL of 0.5 mg/mL in 0.02 M acetic acid) at 37 °C for 2 h. For fibronectin surface coating, slides were treated with human fibronectin solution (100 μL of 10 ug/mL in serum-free media) for 1 h at RT. For fibronectin enriched collagen surface coating, slides were treated with a mixture of bovine Neutralised Type I Collagen Solution (100 μL as above) and fibronectin (4.6 μg) at 37 °C for 2 h. After incubation, all slides were washed with PBS before cell seeding. For 3D matrix culturing, slides were treated with a 1:1 mixture of ice cold Cultrex® Basement Membrane Extract and MEFs in complete media at 37 °C for 20 min. Following incubation, 2% Cultrex®/ DMEM media was added to each well.

## Wound healing migration assay

At approximately 100% confluency, cells were treated with 10 μg/mL Mitomycin C from *S. caespitosus* (#M4287, Sigma-Aldrich). At 3 h post-incubation, monolayers of cells were scratched (vertically) using sterile 20–200 μL tips. Cells were then washed twice with culture medium to remove detached cells and complete culture medium was added to each well containing cells +/− Jasplakinolide (50 nM). Cells were then imaged using Lionheart FX Automated Microscope (Gen5 software v

3.05; BioTek, Agilent Technologies). Migration of cells was captured at 2 h intervals until the complete closure of wound area (37 °C, 5% $CO_2$). Images were subsequently analysed using Fiji using the MRI Wound Healing Tool (RRID:SCR_025260).

## Immunoblotting

Samples were lysed in lysis buffer (1% IGEPAL® CA-630, 10% glycerol, 1% Triton X-100, 150 mM NaCl, 20 mM HEPES pH 7.4, protease inhibitor cocktail tablet (Roche)) and separated by SDS-PAGE using a 4-12% Bis-Tris gel, transferred to PVDF membrane, and blocked with 5% skim milk powder in PBS/0.1% Tween at RT for 1 h. For the analysis of ROCK1 cleavage in MEFs, proteins were immunoblotted using the following antibodies: rabbit anti-ROCK1 (1:1000; Santa Cruz), rabbit anti-cleaved caspase 3 (1:500, Cell Signalling Technology). To demonstrate apoptosis induction proteins were immunoblotted using the following antibodies: rabbit anti-cleaved caspase 3 (1:1000, Cell Signalling Technology), and rabbit anti-PANX1 (1:1000 Santa Cruz). Anti-β actin (AC-15) antibody (Novus Biologicals; NB600-501; 1:4000) was used as a loading control. HRP-conjugated secondary donkey anti-rabbit-HRP (1:5000; Abcam) or Sheep anti-mouse HRP (1:5000; Cytiva), ECL prime detection reagent (Bio-strategy) and ChemiDoc™ Imaging Systems (BioRad) were used to visualise immunoreactive signals. Uncropped immunoblots are presented in Supplementary file.

## Enrichment of FOOD in situ

FOOD was enriched on the cover dish in situ by removing apoptotic cells and other EV subsets in the culture supernatants with a series of PBS washes, before removing remaining cells enzymatically with trypsin/EDTA treatment. Enrichment was validated by quantifying the % remaining cells compared to untreated.

## In-gel digestion

Protein lysates (30 μg) were resolved through electrophoresis in % Bis-Tris Protein Gels (Invitrogen) for 1 h at 150 V and visualised using Coomassie Brilliant Blue stain (0.25% (w/v) Coomassie Blue R-250 (Santa Cruz), 10% (v/v) Acetic acid, 45% (v/v) Methanol) for 1 h. Gels were destained using destain solution (30% (v/v) Methanol, 10% (v/v) Acetic acid) for 3 h. Protein bands were excised from gel lanes using scalpel blades, 10 bands per lane, and destained in destain buffer (25% (v/v) Acetonitrile, 50 mM NH4HCO3) overnight at RT on a shaker. Gel bands were washed with 100% Acetonitrile, and reduced (10 mM Dithiothreitol, 50 mM NH4HCO3) for 45 min at 55 °C (Bio-Rad, Hercules, CA, USA). Gel bands were then alkylated using alkylating solution (55 mM Iodoacetamide, 50 mM NH4HCO3, Sigma) for 30 min in dark before being trypsinised using 5 μg Sequencing Grade Modified Trypsin (Promega), at 37 °C overnight. Digested peptides were then extracted from the gel using 0.1% (v/v) Trifluoroacetic acid (TFA) and 50% (w/v) Acetonitrile on a shaker for 30 min. This step was repeated twice. Extracted peptides were centrifuged (10,000 g for 5 min) and the supernatant was concentrated to a volume of 30 μL using Savant™ SPD131DDA SpeedVac Concentrator (Thermo Scientific).

## LC-MS/MS

Peptide samples were analysed by nano-LC-ESI MS/MS using Orbitrap Eclipse mass spectrometer (Thermo Scientific) equipped with a nanoflow reversed-phase-HPLC (Ultimate 3000 RSLC, Dionex) fitted with an Acclaim Pepmap nano-trap column (Dionex-C18, 100 Å, 75 μm× 2 cm) and an Acclaim Pepmap RSLC analytical column (Dionex-C18, 100 Å, 75 μm × 50 cm). Peptide mix (0.6 μg) was loaded onto the enrichment column at an isocratic flow of 5 μL/min (0.1% (v/v) trifluoroacetic acid, 2% (v/v) acetonitrile) for 5 min prior to switching the enrichment column in-line with the analytical column. The eluents solvent A (5% (v/v) DMSO (Sigma-Aldrich), 0.1% (v/v) formic acid (Sigma-Aldrich)) and solvent B (5% (v/v) DMSO, 100% (v/v) acetonitrile, 0.1% (v/v) formic acid) were flowed at 300 nL/min using a gradient of

90 min at 3–22% B, 10 min at 22–40% B and 5 min at 40–80% B and maintained for 5 min before re-equilibration for 8 min at 3% B prior to subsequent analysis. The mass spectrometer was operated in data-dependent acquisition mode, whereby complete MS1 spectra were acquired in a positive mode at 120,000 resolution. The 'top speed' acquisition mode (3 s cycle time) on the most intense precursor ion was used (i.e., charge states of 2 to 7). MS/MS analyses were performed by 1.6 m/z isolation with the quadrupole, fragmented by HCD with normalised collision energy of 30%. MS2 fragmented ion spectra were acquired at 15,000 resolution. Dynamic exclusion was activated for 30 s and AGC target was set to standard with auto maximum injection mode.

### Database search and protein identification

MaxQuant[68] (v 1.6.17.0) software with integrated Andromeda[69] search engine was used for the identification and quantification of unique proteins, the false discovery rate for both protein and PSM was set to 1% and match between runs enabled. Raw data obtained from tandem mass spectrometry (MS/MS) was searched against UniProt *Homo sapiens* reviewed FASTA sequences. Cysteine carbamidomethylation was set for fixed modification, whereas N-termini acetylation and methionine oxidation were set for variable modifications. Trypsin/P was selected for enzyme specificity to allow maximum of 2 missed cleavages at C-termini of arginine and lysine, also when followed by proline. Peptide mass tolerance for the first and main searches was set to 20 and 4.5 ppm, respectively. Label minimum ratio count of 2 unique or razor peptides was set for quantification. Potential contaminants that may have entered during sample preparation as well as reverse database hits were removed to generate a finalised list of unique proteins. Absolute abundance of each protein was identified using intensity-based absolute quantification (iBAQ), which was calculated by dividing the sum of peak intensities of all proteins by the number of theoretically observable tryptic peptides[70]. Thereon, relative abundance (riBAQ) of each protein was calculated using the following formula: $riBAQ = iBAQ/\Sigma iBAQ$. Proteins with peptide counts less than 2 were disregarded and riBAQ values for each identified protein were represented as percentage (%) for further proteomics analyses.

### Functional enrichment analysis

Funrich[71] (v 3.1.3) software was used for the enrichment analyses of the obtained list of proteins. Statistical significance for plots generated through enrichment analyses were calculated by hypergeometric uncorrected p-value as determined by Funrich[71] software. The built-in analysis tool in FunRich was used to generate the Venn diagram, and figures for gene ontology analysis

### Engulfment of FOOD by LLSM

To monitor FOOD engulfment by LLSM, BMDMs were first generated as specified above. The day prior to imaging, 10,000 MEFs per well were seeded in an ibidi μ-Slide 8 Well Glass Bottom chamber slide. The following day, MEFs were treated with an apoptotic cocktail (10 μM S63845 and 5 μM ABT-737) for 4 h. FOOD/F-ApoEVs were then enriched in situ as described above. BMDMs were stained with CTV (1.8 μL CTV + 10 mL serum free media, 20 min, 37 °C, 10% CO₂) and washed. Approximately 40,000 CTV-labelled BMDMs were then added to the isolated FOOD and imaged by LLSM for 2-4 h with 2 min intervals at 37 °C, 5% CO₂. Time-lapse live-cell data were acquired using a Lattice Light Sheet 7 (Zeiss). Light sheets (488 nm, 561 nm and 640 nm) of 30 μm in length with a thickness of 1 μm were created at the sample plane via a 13.3 × 0.44 numerical aperture (NA) objective. Fluorescence emission was collected via a 44.83 × 1 NA detection objective via a multiband stop, LBF 405/488/561/633, filter. Aberration correction was set to a value of 170 to minimize aberrations as determined by imaging the Point Spread Function using 100 nm fluorescent microspheres. Data were collected with a frame time of 5 ms and a z step of 0.4 μm

with 834 frames per 290 μm by 290 μm. Data were subsequently deskewed and then deconvolved using a constrained iterative algorithm with 15 iterations.

### Re-feeding engulfment assay

To monitor subsequent engulfment following FOOD uptake by BMDMs, either $1 \times 10^4$ or $2 \times 10^4$ MEFs were seeded into a 48 well plate and induced to undergo apoptosis. FOOD/F-ApoEVs were then enriched in situ as described above. CTV-labelled BMDMs were then added to the FOOD-enriched well or an untreated well and incubated for 24 h. Apoptotic CypHer5E labelled Jurkat T cells (induced to undergo apoptosis with 0.5 μM S63845, 5 μM ABT-737) were then added to the BMDMs at a 1:5 ratio for 1 h, and engulfment was assessed by flow cytometry.

### Isolation of F-ApoEVs and fluorescence activated cell sorting (FACS)

To isolate F-ApoEVs, cells were first seeded on tissue culture plates coated with bovine Neutralised Type I Collagen Solution (0.5 mg/mL, as described above) and apoptosis induced with 0.5 μM S63845 + 5 μM ABT-737 or infection with IAV, as above. ApoBDs, apoptotic cells and other EV subsets in the culture supernatants were removed with a series of PBS washes. Next, the collagen coating was enzymatically digested with collagenase type 1 (0.5 mg/mL) for 30 min at 37 °C on a shaking incubator. The digested collagen solution was collected and centrifuged at 500 x g for 10 min to pellet remaining cells, then 3000 x g for 20 min to pellet F-ApoEVs. F-ApoEVs were next enriched using MACS cell separation using an Annexin 5 Microbead Kit (Miltenyi Biotec) according to the manufacturers protocol. Enriched F-ApoEV samples were resuspended in FACS buffer (10% (v/v) FCS, PBS, 1 x A5-BB), stained with A5 and filtered through a 70 μm cell strainer prior to sorting. F-ApoEVs were identified as A5⁺, and FSC^low, and sorted using a BD FACSAria Fusion. Notably, all F-ApoEV samples were centrifuged at 3000 x g as this is sufficient to pellet F-ApoEVs; however free virions are likely to remain in the discarded supernatant.

### Transmission electron microscopy

F-ApoEVs were isolated as above and fixed in 2.5% glutaraldehyde/PBS at 4 °C 1 h, then washed with Sorensen's phosphate buffer (PB). Pellets were post-fixed in 1% $OsO_4$, 1.5% Potassium ferrocyanide for 2 h, 4 °C. Staining with 2.5% uranyl acetate followed overnight, 4 °C. Subsequently, pellets were rinsed with PB then dehydrated in an ethanol gradient (20%, 50%, 70%, 90%, 100%, 100%) for 1 h followed by acetone (100%, 100%). Pellets were next embedded in Procure 812 resin using microwave-assisted infiltration (25%, 50%, 75%, 100%, 100%; at 250 W under vacuum for 3 min). Subsequently, pellets were transferred to 100% epon and embedded in a silicone mould for polymerisation for 48 h at 60 °C. Resin blocks were then sectioned using a DiATOME 45° Ultra Diamond Knife on a Leica EM UC7-ultramicrotome, and 70 nm thickness sections were obtained and mounted on EM-copper grids with formvar/carbon coating. Sections were poststained in 4% UAc in water and lead citrate for 5 min each, and then processed for imaging using a Jeol JEM-2100 TEM at 80 kV using an AMT NanoSprint II CMOS camera.

### F-ApoEV-A549 co-incubation assays

F-ApoEV-A549 co-incubation assays were performed by incubating $2.5 \times 10^3$, $2 \times 10^4$, or $4 \times 10^4$ FACs isolated F-ApoEVs with $2 \times 10^4$ target cells (A549 cells) for 48 h at 37 °C in 5% CO₂, unless otherwise specified. Target cells were then stained by A5-V450 and Live Dead-APC, fixed, permeabilised, and stained with NP for flow cytometry analysis.

### Statistics and reproducibility

Raw data were visualized and processed using Microsoft Excel (v 2405) and GraphPad Prism (v 10, GraphPad Software Inc.). Where specified,

statistical significance was determined by performing an unpaired two-tailed Student's *t*-test. No statistical method was used to predetermine sample size. No data were excluded from analyses. The experiments were not randomized. The Investigators were not blinded to allocation during experiments and outcome assessments.

## Reporting summary

Further information on research design is available in the Nature Portfolio Reporting Summary linked to this article.

## Data availability

All data associated with this study are present in the main text or the Supplementary Information. Proteomic data are available in Supplementary Data 1 and raw data available on ProteomeXchange with identifier PXD067718. Source data are provided with this paper.

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

## Acknowledgements

The authors would like to thank the Margaret Veale, Chad Johnson, Sarah Wilson, and Kristian Caracciolo from the La Trobe University Bioimaging Platform for their technical support. This work was supported by a La Trobe University Postgraduate scholarship to S.F.R., the National Health and Medical Research Council (2009287 to GAS, 1140187 to IP), Australian Research Council (230101056 and 210102658 to IP), Jack Brockhoff Foundation (4852 to GAS) and L'Oreal UNESCO For Women in Science (to GAS). This work was completed on Wurundjeri Woi Wurrung country. The authors would like the acknowledge the Traditional Custodians of this Country and pay our respect to Elders past and present.

## Author contributions

S.F.R performed all experiments with help from T.K., G.F.R., B.S., C.L.V., D.C.O., A.L.H., G.A.S., and I.K.H.P. S.F.R., G.A.S, and I.K.H.P. designed all experiments. S.W.C. and K.B. provided ECM coating expertise. T.K., P.F., C.A., and S.M., aided with proteomics. M.O. designed and generated ROCK1nc mice. J.R., N.D.G., K.L.R. provided microscopy support. P.R. provided bioimage analysis support. S.F.R. wrote the manuscript with help from G.A.S., I.K.H.P., and input from all authors.

## Competing interests

The authors declare no competing interests.
