## [Transparent Peer Review file · Nature Communications]

The formation of the 'footprint of death' as a mechanism for generating large substrate-bound extracellular vesicles that mark the site of cell death

Corresponding Author: Professor Ivan Poon

Version 0:

Reviewer comments:

Reviewer #1

(Remarks to the Author)

In this study, the authors discover that apoptotic cells can contract and form a 'footprint of death' (FOOD), which is a membrane-enclosed, F-actin-rich structure tightly anchored to the substrate, marking the site of cell death. They convincingly demonstrate that this represents a novel mechanism for large extracellular vesicle (EV) generation by apoptotic cells. Furthermore, the authors show that the formation of FOOD depends on ROCK1 and propose its potential physiological roles in efferocytosis and viral infection. Overall, the findings present an interesting and potentially impactful story, and the data are generally convincing. I have a few suggestions to further improve the manuscript:

1. Given the striking morphological similarity between FOOD and migrasomes, and considering that ROCK1 is an essential gene for migrasome formation, both processes involve contraction or migration, which generates mechanical forces to pull membrane tubules. Do FOOD and migrasomes share similar mechanisms and broadly represent related phenomena? The authors may want to elaborate on this point in the discussion.
2. It is important to note that migrasome formation does not necessarily require migration. Recent studies show that bacterial toxins can induce non-canonical migracytosis and migrasome formation without migration. The authors should cite and discuss this study.
3. Along these lines, does chemical inhibitors known to block migrasome formation, such as inhibitors of PIP5KA or SMS2, that could also inhibit FOOD formation?
4. The manuscript highlights two functions of F-ApoEVs: enhancing macrophage phagocytic capacity and serving as carriers for viral spread. Are these functions unique to F-ApoEVs? Do apoptotic cell bodies or other EVs generated during apoptosis (e.g., ApoBDs) exhibit similar effects?
5. FOOD can form on glass cover slides or surfaces coated with extracellular matrix (ECM). It would be important to statistically analyze whether the presence of ECM affects the quantity of FOOD formation, as this could provide valuable insights into the role of ECM in the process.
6. The mass spectrometry results indicate histones are enriched in FOOD, yet subsequent validation shows that DNA and histones are absent in both FOOD and F-ApoEVs. The reliability of the isolation protocol, as shown in Supplementary Figure 6a, warrants further validation.

Reviewer #2

(Remarks to the Author)

To support the claim that F-actin is present in FOOD, the authors performed CLEM, whereby they correlated Confocal Laser Scanning and Scanning electron microscopy images of a MEF (Figure 3f). While some details are different on these images, such as the lack of the structures in SEM corresponding to the bright spot located just right of the nucleus and one of thin filaments seen in the lower part of the insets, overall these images showed a consistent morphology. I suggest adding the following information to the manuscript:

- The time interval between recording CLSM images and the fixation required for SEM

- Please state what might be the reason for the small differences between CLSM and SEM images mentioned above
- Please explain how was the cell imaged in CLSM located in SEM (the size of the CLSM tile map, how was the overall region found in SEM, what was used to directly identify the same cell in CLSM and SEM images, ...)
- The dashed squares on the CLSM images, especially on the middle image, are not positioned precisely

Reviewer #3

(Remarks to the Author)

The authors of this manuscript elegantly describe, using various microscopy techniques, a new phenomenon observed during apoptosis. This phenomenon ultimately leads to the generation of a type of vesicle distinct from apoptotic bodies, which promotes increased efferocytosis. However, several experiments are necessary to strengthen the validity of their claims.

Major concerns

- My primary concern is regarding how F-ApoEVs differ from the generation of migrasomes described previously in the literature. Indeed, F-ApoEVs share many similar characteristics with migrasomes, including comparable average size, a round shape, localization at the terminal or cross connections of retraction fibers, and difficulty in displacement. Several proteins also serve as markers for migrasomes, including an enrichment of cytoskeletal proteins. Furthermore, it has been shown that the formation of migrasomes can be dependent on ROCK1. (doi.org/10.1038/s41421-020-0179-6). The ability of F-ApoEVs to act as viral reservoirs and participate in the propagation of infection is another characteristic similar to what has been reported for migrasomes (doi: 10.1016/j.virs.2023.06.001). Consequently, it appears F-ApoEVs exhibit numerous features of migrasomes. Since the authors claim that FOOD represents a new mechanism and a new type of extracellular vesicle (EV), it is essential to validate the uniqueness of F-ApoEVs rather than categorizing them as migrasomes secreted by apoptotic cells. A proteomic analysis comparing migrasomes from non-apoptotic cells to F-ApoEVs would be a way to demonstrate this distinction. Currently, the only result presented by the authors to differentiate F-ApoEVs from migrasomes is the use of Jasplakinolide to block migration. This is insufficient to justify the novelty claimed in this manuscript. Overall, while this manuscript presents interesting data, it requires additional controls and experiments to meet the level of novelty asserted here and to convince that F-ApoEVs are not merely migrasomes secreted by apoptotic cells.

Several structures shown are similar to filopodia/retraction fibers observed in the formation of migrasomes. Since the authors claim that this represents a new mechanism, they must demonstrate that the phenomenon observed under apoptotic conditions is absent under non-apoptotic conditions or, if not, demonstrate that molecular regulators and/or effectors of apoptosis (activated caspase-9 or caspase-3 for example) are involved in their biogenesis. However, no controls are provided alongside the various experimental conditions. In Figure 1F, the percentage of FOOD formation is measured, reaching between 80% and 100%. It is essential to quantify the level of apoptosis achieved in each condition to accurately assess the actual frequency of FOOD formation. For example, if only 50% of the cells are apoptotic, how can it be explained that 100% of the cells exhibit these structures?

Figure 1G. Cells can normally exhibit "branch" structures. The authors should compare the measured characteristics to a non-apoptotic control (the vehicle used to solubilize the cocktail).

The authors demonstrate the induction of apoptosis through caspase-3 cleavage in A549 cells. Given that the entire premise lies on the assumption that the cells indeed undergo apoptosis, it is crucial to thoroughly characterize the apoptotic state of the cells. Are all the cells in an apoptotic state, or are some in a normal state? The primary treatment used to describe FOOD and F-ApoEVs is a BH3 mimetic cocktail. It is necessary to explain the mechanism by which the BH3 mimetic cocktail induces apoptosis and to confirm that the cells respond as expected. Does the BH3 cocktail treatment also induce other forms of cell death like necroptosis or an autophagic state in the cells used? The authors should also demonstrate that other treatments, like the BH3 cocktail, indeed induce characteristics of apoptosis, such as caspase-3 cleavage. Overall, it is essential to thoroughly characterize their apoptotic models.

- Figure 2G. The authors use Jasplakinolide to inhibit migration. However, there remains a significant migratory capacity, as presented in supplementary figure 5. Consequently, there should be a quantification of the number/diameter of F-ApoEVs secreted by cells treated with Jasplakinolide compared to the vehicle, similar to how it was quantified in figures 2C and D. The data presented to justify that F-ApoEVs are not migrasomes is weak and does not allow for their exclusion.

Minor concerns

- In the supplementary figure 2, Why did the authors choose not to use PKH26 to stain the cell membrane, as was done in Figure 1 and Supplementary Figure 1?
- Define all abbreviations in the legends (CLSM, ESID, SEM, etc.). They are defined in the text, but not in the respective legend.

Version 1:

Reviewer comments:

Reviewer #1

(Remarks to the Author)

The authors have addressed my queries satisfactorily. Congratulations on this elegant work!

Reviewer #2

(Remarks to the Author)

The authors have successfully answered all points I raised in the previous round. There are only few minor issues.

1. (Lines 102-106, Supplementary figure 3): Please state what is PANX1, and explain the relevance of the PANX1 cleavage for the data shown.
2. (Supplementary figure 7, legend): Words “nucleus” and “exposed phosphatidylserine” should be interchanged.
3. (Lines 550-554, Supplementary Figure 12): Please provide more information the functional enrichment analysis. This includes explaining what are the p-values shown on this figure, and stating the statistical hypothesis tested and the parameters of the distribution.
4. Spelling errors that changed the meaning of words: “thought” (line 284), “nothing” (line 331).

Reviewer #3

(Remarks to the Author)

The authors addressed past criticism and provided convincing additional results.

Dear Dr Morales,

Thank you very much for considering our manuscript at *Nature Communications*. We are pleased that the Reviewers gave positive feedback on our work and we welcome their constructive critiques. Please see below point-by-point response addressing all Reviewer questions and concerns. As requested, we have now performed new experiments (Supplementary Figure 3, 4, 6, 7, 9 and 10), which are included in the revised manuscript and have clarified any misinterpretations. Revised text is highlighted in red in the main manuscript, including the inclusion of one additional author.

Reviewer #1:

In this study, the authors discover that apoptotic cells can contract and form a 'footprint of death' (FOOD), which is a membrane-enclosed, F-actin-rich structure tightly anchored to the substrate, marking the site of cell death. They convincingly demonstrate that this represents a novel mechanism for large extracellular vesicle (EV) generation by apoptotic cells. Furthermore, the authors show that the formation of FOOD depends on ROCK1 and propose its potential physiological roles in efferocytosis and viral infection. Overall, the findings present an interesting and potentially impactful story, and the data are generally convincing. I have a few suggestions to further improve the manuscript:

1. Given the striking morphological similarity between FOOD and migrasomes, and considering that ROCK1 is an essential gene for migrasome formation, both processes involve contraction or migration, which generates mechanical forces to pull membrane tubules. Do FOOD and migrasomes share similar mechanisms and broadly represent related phenomena? The authors may want to elaborate on this point in the discussion.

Response: We would like to thank Reviewer #1 for their time and input into our manuscript. Similar to Lu *et al.* 2020 (PMID:32802402) who reported the importance of ROCK1 in the formation of migrasomes, here, we demonstrated that the specific activation of ROCK1 by caspase-mediate cleavage during apoptosis is important for FOOD formation. As recommended by the Reviewer, we have now extended our discussion to elaborate on the morphologic and mechanistic similarities between FOOD and migrasome formation (please see lines 325-334). Notably, we have also performed additional experiments (e.g. extended time course studies, migrasome inhibitor studies) to further distinguish FOOD from the formation of migrasomes (please see Supplementary Figure 7, 9, 10 and Response #3 below).

2. It is important to note that migrasome formation does not necessarily require migration. Recent studies show that bacterial toxins can induce non-canonical migracytosis and migrasome formation without migration. The authors should cite and discuss this study.

Response: We have now highlighted the findings of this publication on non-canonical migracytosis (PMID:39500876) in our discussion (please see lines 325-331). Please note that

a key distinction between FOOD formation and non-canonical migracytosis is that non-canonical migracytosis occurs in the absence of death (PMID:39500876).

3. Along these lines, does chemical inhibitors known to block migrasome formation, such as inhibitors of PIP5KA or SMS2, that could also inhibit FOOD formation?

Response: We thank the Reviewer for these suggestions. As recommended, we have now performed new experiments to assess the formation of FOOD/F-ApoEVs in the presence of the PIP5K α inhibitor (ISA-2011B; 20 μ M) and SMS2 inhibitor (SMS2-IN-1; 30 μ M). As shown in the new Supplementary Figure 10 (and below), treatment of apoptotic MEFs with SMS2-IN-1 or ISA-2011B did not impair the formation of FOOD. Moreover, quantification of microscopy data demonstrates that inhibitors of migrasomes formation did not induce a noteworthy change in either the number or size of F-ApoBDs. We have highlighted these new findings in the revised manuscript (lines 144-148).

Supplementary Figure 10: Pharmacological inhibitors of migrasome formation do not inhibit F-ApoEV formation. **a** Confocal laser scanning microscopy (CLSM) imaging of apoptotic MEFs treated with a BH3-mimetic cocktail (5 μ M ABT-737, 10 μ M S63845) with and without migrasome inhibitors SMS2-IN-1 (30 μ M) and, ISA-2011B (20 μ M). MEFs were stained with SiR-actin, A5-FITC, and Hoechst 33342, to visualise F-actin, the nucleus, and exposed phosphatidylserine, respectively. Quantification of the number of F-ApoEVs generated per cell (**b**), and the diameter of F-ApoEVs (μ m) (**c**) generated by apoptotic MEFs alone, or in the presence of migrasome inhibitor SMS2-IN-1(30 μ M), or ISA-2011B (20 μ M). Data points in (**b**) represent individual cells (n=23, n=19, and n=27, respectively) pooled from (n=3) independent experiments. Data points in (**c**) represent individual F-ApoEVs (n=789, n=562 and n=750) pooled from (n=3) independent experiments. Solid red line indicated mean, dashed red line indicates quarterlies. Unpaired student's two tailed t-test was performed to determine the indicated p-values.

4. The manuscript highlights two functions of F-ApoEVs: enhancing macrophage phagocytic capacity and serving as carriers for viral spread. Are these functions unique to F-ApoEVs? Do apoptotic cell bodies or other EVs generated during apoptosis (e.g., ApoBDs) exhibit similar effects?

Response: Our team and others have previously demonstrated that ApoBDs generated from virus-infected apoptotic cells can propagate influenza A virus (PMID:32385344) and African swine fever virus infections (PMID:37983498). In addition to viral infection, the engulfment of apoptotic cells can promote subsequent rounds of efferocytosis (PMID:32004476). In this manuscript, our results demonstrate that the formation of F-ApoEVs represents another mechanism to aid viral propagation and efferocytosis following apoptosis. The generation of multiple subsets of apoptotic cell-derived EVs which exhibit similar functional roles represents an effective multipronged approach to ensure downstream effects. We have now included these studies and concepts in the discussion (lines 283-285, 301-304).

5. FOOD can form on glass cover slides or surfaces coated with extracellular matrix (ECM). It would be important to statistically analyze whether the presence of ECM affects the quantity of FOOD formation, as this could provide valuable insights into the role of ECM in the process.

Response: We thank the reviewer for this suggestion to improve our manuscript. We have now quantified data presented in Figure 1h and 1i examining the formation of FOOD on various ECM substrates. As shown in new Supplementary Figure 6 (and below), comparable levels and size of F-ApoEVs were formed on various ECM substrates.

Supplementary Figure 6: FOOD/ F-ApoEVs readily form on surfaces coated with ECM proteins. **a** Quantification of the number of F-ApoEVs generated per cell from confocal laser scanning microscopy (CLSM) imaging of apoptotic MEFs on uncoated, collagen, fibronectin, and fibronectin and collagen coated chamber slides, treated with a BH3 mimetic cocktail (5 μM ABT-737, 10 μM S63845). Data points represent individual cells (n=21, n=25, n=30 and n=19, respectively) pooled from (n=3) independent experiments. **b** Quantification of F-ApoEV diameter (μm) from CLSM imaging of apoptotic MEFs uncoated, collagen, fibronectin, and fibronectin and collagen coated chamber slides, as previous. Data points represent individual F-ApoEVs (n=952, n=1882, n=2640, and n=1502, respectively) pooled from (n=3) independent experiments. Solid red line indicated mean, dashed red line indicates quarterlies. Unpaired student's two tailed t-test was performed to determine the indicated p-values.

6. The mass spectrometry results indicate histones are enriched in FOOD, yet subsequent validation shows that DNA and histones are absent in both FOOD and F-ApoEVs. The reliability of the isolation protocol, as shown in Supplementary Figure 6a, warrants further validation.

Response: As highlighted by the Reviewer, our proteomics analysis indicated that histones and other nuclear material were enriched in F-ApoEVs. However, upon further analysis by confocal microscopy, we confirmed that FOOD did not contain nuclear material. The nuclear material detected by proteomics likely represents components released from the cell during apoptosis and possibly secondary necrosis, as mentioned in line 190-193. Nevertheless, the Reviewer raised an important point and we have now included additional data acquired during the development of the isolation protocol. As shown in the new Supplementary Figure 11 (microscopy data and associated quantification) and below, trypsin/EDTA treatment robustly and consistently removed whole cells from the culture dish and enriched FOOD/F-ApoEVs. We hope these data will satisfy the Reviewer's concern.

Supplementary Figure 11: In situ isolation of FOOD. **a** Schematic diagram of in situ FOOD isolation approach. **b** Representative confocal laser scanning microscopy images of untreated cells and FOOD post-treatment with trypsin/EDTA. **i**, **ii**, and **iii** indicate data generated from three independent experiments. **c** Quantification of remaining cells post FOOD isolation, determined by the number of cells remaining after isolation divided by the number of cells present in the untreated sample (n=6). Cells were stained with PKH26, and A5-FITC and Hoechst 33342. Error bars represent s.e.m. Unpaired student's two tailed t-test was performed to determine the indicated p values.

Reviewer #2 (Remarks to the Author):

To support the claim that F-actin is present in FOOD, the authors performed CLEM, whereby they correlated Confocal laser scanning and Scanning electron microscopy images of a MEF (Figure 3f). While some details are different on these images, such as the lack of the structures in SEM corresponding to the bright spot located just right of the nucleus and one of thin filaments seen in the lower part of the insets, overall these images showed a consistent morphology. I suggest adding the following information to the manuscript:

Response: We would like to thank the Reviewer for providing their expertise and insight into our manuscript. We greatly appreciate their time in assessing our CLSM approach and we have made all the required changes in the revised manuscript (please see lines 437-455 and below).

1. The time interval between recording CLSM images and the fixation required for SEM.

Response: Samples were fixed prior to confocal imaging. Thus, CLSM and SEM images were taken at the same timepoint (lines 438-444).

2. Please state what might be the reason for the small differences between CLSM and SEM images mentioned above.

Response: The small differences between CLSM and SEM images is likely due to repeated buffer exchanges during the SEM processing procedure. We have now highlighted this in the revised Materials and Methods (lines 452-454).

3. Please explain how was the cell imaged in CLSM located in SEM (the size of the CLSM tile map, how was the overall region found in SEM, what was used to directly identify the same cell in CLSM and SEM images, ...).

Response: We have revised the Materials and Methods accordingly to describe how cells imaged in CLSM were located in SEM (lines 444-452).

4. The dashed squares on the CLSM images, especially on the middle image, are not positioned precisely.

Response: We have adjusted Figure 3f accordingly in the revised manuscript.

Reviewer #3 (Remarks to the Author):

The authors of this manuscript elegantly describe, using various microscopy techniques, a new phenomenon observed during apoptosis. This phenomenon ultimately leads to the generation of a type of vesicle distinct from apoptotic bodies, which promotes increased efferocytosis. However, several experiments are necessary to strengthen the validity of their claims.

Major concerns:

1. My primary concern is regarding how F-ApoEVs differ from the generation of migrasomes described previously in the literature. Indeed, F-ApoEVs share many similar characteristics with migrasomes, including comparable average size, a round shape, localization at the terminal or cross connections of retraction fibers, and difficulty in displacement. Several proteins also serve as markers for migrasomes, including an enrichment of cytoskeletal proteins. Furthermore, it has been shown that the formation of migrasomes can be dependent on ROCK1. (doi.org/10.1038/s41421-020-0179-6). The ability of F-ApoEVs to act as viral reservoirs and participate in the propagation of infection is another characteristic similar to what has been reported for migrasomes ([doi: 10.1016/j.virs.2023.06.001](https://doi.org/10.1016/j.virs.2023.06.001)). Consequently, it appears F-ApoEVs exhibit numerous features of migrasomes. Since the authors claim that FOOD represents a new mechanism and a new type of extracellular vesicle (EV), it is essential to validate the uniqueness of F-ApoEVs rather than categorizing them as migrasomes secreted by apoptotic cells. A proteomic analysis comparing migrasomes from non-apoptotic cells to F-ApoEVs would be a way to demonstrate this distinction. Currently, the only result presented by the authors to differentiate F-ApoEVs from migrasomes is the use of Jasplakinolide to block migration. This is insufficient to justify the novelty claimed in this manuscript. Overall, while this manuscript presents interesting data, it requires additional controls and experiments to meet the level of novelty asserted here and to convince that F-ApoEVs are not merely migrasomes secreted by apoptotic cells.

Response: We would like to thank the Reviewer for the comprehensive assessment of our manuscript. We are extremely appreciative of their time and input. The Reviewer has highlighted several overlapping characteristics between FOOD/F-ApoEVs and migrasomes including morphologic and mechanistic similarities, as well as functional roles. As highlighted in the manuscript, we also noted the striking similarities between the two phenomena. However, our data demonstrates that the formation of FOOD is a distinct cellular process from migrasome formation. This is based on the following findings:

- (i) Time-lapse microscopy data presented in Figure 2f demonstrates that A431 cells do not migrate and move outside their initial cell boundaries but are still capable of generating FOOD.
- (ii) We have performed new long-term imaging experiments and showed that A5+ large EVs are not formed prior to apoptosis induction (please see new Supplementary Figure 7).

- (iii) To ascertain the importance of cell death rather than cell migration is required for FOOD/F-ApoEV formation, we performed additional experiments with Bax^{-/-}Bak^{-/-} cells. These cells lack the key pro-apoptotic proteins Bax and Bak and are unable to undergo intrinsic apoptosis and did not generate FOOD/F-ApoEVs following BH3-mimetics treatment. These new data demonstrate that apoptosis is required for FOOD/F-ApoEV formation (please see new Supplementary Figure 4).
- (iv) Inhibition of cell migration through Jasplakinolide treatment, as shown in Figure 2 and new quantification presented in Supplementary Figure 9, does not impair the formation of FOOD/F-ApoEV formation.
- (v) We have performed new experiments to specifically examine whether inhibitors of migrasome formation alter the generation of FOOD/F-ApoEVs. This includes inhibitors targeting PIP5K α and SMS2 (PMID:37142675, 37488437). As shown in the new Supplementary Figure 10, inhibitors of migrasome formation do not inhibit the formation of FOOD/F-ApoEVs.
- (vi) We have examined the publicly available proteomic dataset on migrasomes derived from NKR epithelial cells published in Zhao et al. 2022 (PMID:31123599). Key migrasome markers identified via proteomics in PMID:31123599 are not enriched in FOOD. We have highlighted this in the revised discussion (lines 331-334).

2. Several structures shown are similar to filopodia/retraction fibers observed in the formation of migrasomes. Since the authors claim that this represents a new mechanism, they must demonstrate that the phenomenon observed under apoptotic conditions is absent under non-apoptotic conditions or, if not, demonstrate that molecular regulators and/or effectors of apoptosis (activated caspase-9 or caspase-3 for example) are involved in their biogenesis. However, no controls are provided alongside the various experimental conditions. In Figure 1F, the percentage of FOOD formation is measured, reaching between 80% and 100%. It is essential to quantify the level of apoptosis achieved in each condition to accurately assess the actual frequency of FOOD formation. For example, if only 50% of the cells are apoptotic, how can it be explained that 100% of the cells exhibit these structures?

Response: As requested by the Reviewer, we have now performed additional analysis to further validate that the formation of FOOD occurs specifically under apoptotic conditions. This includes the following new experiments:

- i) Apoptosis validation by immunoblotting:** To validate that the BH3-mimetic cocktail is specifically inducing apoptosis, we analysed the molecular regulators of apoptosis via immunoblotting. We demonstrate that ABT-737/S63845 treatment results in apoptotic hallmarks such as cleavage of caspase 3 and Pannexin 1 (PMID:31069027). We have also performed similar experiments under other apoptotic conditions as described in our manuscript. Please find this new data in Supplementary Figure 3.
- ii) Apoptosis validation by flow cytometry:** We further demonstrate that AB7-737/S63845 treatment results in apoptosis by examining cell viability by flow

cytometry analysis. Quantification of flow cytometry data demonstrates that ~85% of MEFs undergo caspase-dependent cell death upon treatment with the BH3-mimetic cocktail for 4 hours. Please find this new data in Supplementary Figure 3 (and below).

iii) FOOD formation occurs only after apoptosis induction: In a new experiment now presented in Supplementary Figure 7 (and below), we performed time-lapse imaging of MEFs for 4 hours before adding the apoptosis inducers ABT-737/S63845, and imaging for a further 4 hours. This new analysis demonstrates that FOOD/F-ApoEVs are only generated upon the induction of apoptosis.

iv) Genetically limiting apoptosis prevents FOOD formation: Finally, we utilised a genetic approach by utilising Bax/Bak double knock out ($Bax^{-/-}Bak^{-/-}$) MEFs which are unable to undergo mitochondria-mediated (intrinsic) apoptosis (PMID:8358790). As expected, $Bax^{-/-}Bak^{-/-}$ MEFs cells displayed no morphological hallmarks of apoptosis upon treatment with the BH3-mimetic cocktail and did not form FOOD. Please find this new data in Supplementary Figure 4 (and below).

v) Well-characterised properties of BH3-mimetics: Please see Response #4 below regarding the mechanism of action of ABT-737 and S63845.

Taken together, we provided substantial evidence that FOOD formation is apoptosis-specific. We have also highlighted this in the main text (lines 102-106).

Supplementary Figure 3: Validation of cell death inducing stimuli. The reduction in cell viability was determined by flow cytometry (as measure by phosphatidyl serine exposure using A5-FITC staining) and apoptosis was confirmed by immunoblot analysis of the following: (a,b) MEFs 4 h post treatment with BH3-mimetic cocktail (5 μ M ABT-737, 10 μ M S63845), (c,d) MEFs 24 h post UV-irradiation (150 mJ/cm²), (e,f) MEFs 24 h post etoposide (125 nM), and (g,h) A549 cells 24 h post infection with IAV (MOI=10), with or without the presence of pan-caspase inhibitor Q-VD-OPh (Q-VD) (50 μ M) to inhibit apoptosis. Asterisk indicates a non-specific band which occurs due the induction of apoptosis and co-treatment with Q-VD-OPh⁷¹. Error bars represent s.e.m. At least three independent experiments were performed for all experiments unless otherwise specified.

Supplementary Figure 4: The loss of Bax/Bak prevents the formation of FOOD/F-ApoEVs. WT or $Bax^{-/-} Bak^{-/-}$ MEFs were treated with a BH3-mimetic cocktail (5 μ M ABT-737, 10 μ M S63845) and imaged by time lapse confocal laser scanning microscopy (CLSM) to monitor the formation of FOOD. F-actin and the nucleus were visualised by SiR-actin and Hoechst 33342 staining, respectively, and exposed phosphatidyserine with A5-FITC. At least three independent experiments were performed for all experiments unless otherwise specified.

Supplementary Figure 7: FOOD/ F-ApoEV formation does not occur prior to apoptosis induction. MEFs were stained with SiR-actin, A5-FITC, and Hoechst 33342, to visualise F-actin, nucleus, and exposed phosphatidyserine, respectively, and imaged via confocal laser scanning microscopy for 200 min. Imaging was paused and MEFs were then treated with a BH3-mimetic cocktail (5 μ M ABT-737, 10 μ M S63845) to induce apoptosis, and imaged for an additional 210 min. At least three independent experiments were performed for all experiments unless otherwise specified.

We would like to clarify that for the experiments quantifying FOOD formation, this was achieved by calculating the percentage of FOOD forming cells from total apoptotic cells (annexin V positive cells displaying apoptotic morphologies). As noted above, the BH3-mimetic cocktail induces ~85% apoptosis. To avoid confusion, we have clarified this quantification in the figure legend of the revised manuscript (lines 802-805).

3. Figure 1G. Cells can normally exhibit "branch" structures. The authors should compare the measured characteristics to a non-apoptotic control (the vehicle used to solubilize the cocktail).

Response: As mentioned above in response to point 2, the formation of FOOD is specific to apoptosis and is identified by characteristics such as A5 staining. As observed in the new Supplementary Figure 4 and Figure 7, A5+ FOOD-like branch structures are not observed under non-apoptotic conditions.

4. The authors demonstrate the induction of apoptosis through caspase-3 cleavage in A549 cells. Given that the entire premise lies on the assumption that the cells indeed undergo apoptosis, it is crucial to thoroughly characterize the apoptotic state of the cells. Are all the cells in an apoptotic state, or are some in a normal state? The primary treatment used to describe FOOD and F-ApoEVs is a BH3 mimetic cocktail. It is necessary to explain the mechanism by which the BH3 mimetic cocktail induces apoptosis and to confirm that the cells respond as expected. Does the BH3 cocktail treatment also induce other forms of cell death like necroptosis or an autophagic state in the cells used? The authors should also demonstrate that other treatments, like the BH3 cocktail, indeed induce characteristics of apoptosis, such as caspase-3 cleavage. Overall, it is essential to thoroughly characterize their apoptotic models.

Response: Please see response to point 2 above related to the confirmation of apoptosis induction. We would like to clarify that the BH3-mimetic cocktail routinely used to induce apoptosis in this study is a gold-standard approach for intrinsic apoptosis induction in the cell death field. This cocktail includes:

- ABT-737 = a small molecule inhibitor that targets the pro-survival proteins Bcl-2 and Bcl-xL with high selectivity and affinity. Binding of ABT-737 to Bcl-2/Bcl-xL prevents their interaction with pro-apoptotic protein Bax (PMID:17460700).
- S63845 = selective inhibitor that specifically binds to the BH3-binding groove of the pro-survival protein MCL-1 (PMID:27760111).

Binding of these chemical inhibitors to their targets sequesters their pro-survival functions and unleashes mitochondrial-mediated apoptosis. As both inhibitors are highly selective for their targets, there are no off-target effects that trigger alternative forms of cell death (such as necroptosis). They represent the most precise method to induce intrinsic apoptosis. As requested, we have now included these details in the manuscript. Please see lines 364-367.

5. Figure 2G. The authors use Jasplakinolide to inhibit migration. However, there remains a significant migratory capacity, as presented in supplementary figure 5. Consequently, there should be a quantification of the number/diameter of F-ApoEVs secreted by cells treated with Jasplakinolide compared to the vehicle, similar to how it was quantified in figures 2C and D. The data presented to justify that F-ApoEVs are not migrasomes is weak and does not allow for their exclusion.

Response: Please see response to point 1 above in regard to our findings that demonstrate F-ApoEV formation is clearly distinct to migrasome formation. In addition, as requested by the Reviewer, we have now performed new data analysis to compare the F-ApoEVs generated in the presence or absence of the migration inhibitor, Jasplakinolide. This new analysis demonstrates that Jasplakinolide treatment did not impair the formation of F-ApoEVs and did not induce a noteworthy impact on the size of F-ApoEVs. Please see data in the new Supplementary Figure 9 (and below).

Supplementary Figure 9: Pharmacological inhibition of cell migration by jasplakinolide does not prevent the formation of FOOD. **a** Quantification of the number of F-ApoEVs generated per cell from confocal laser scanning microscopy (CLSM) imaging of apoptotic MEFs treated with a BH3-mimetic cocktail (5 µM ABT-737, 10 µM S63845) alone, or with migration inhibitor Jasplakinolide (50 nM). Data points represent individual cells (n=34 and n=27, respectively) pooled from (n=3) independent experiments. **b** Quantification of the diameter (µm) of F-ApoEVs from CLSM imaging of apoptotic MEFs with and without Jasplakinolide treatment (50 nM). Data points represent individual F-ApoEVs (n=1625 and n=1014) pooled from (n=3) independent experiments. Solid red line indicates mean, dashed red line indicates quarterlies. Unpaired student's two tailed t-test was performed to determine the indicated p-values.

Minor concerns

6. In the supplementary figure 2, Why did the authors choose not to use PKH26 to stain the cell membrane, as was done in Figure 1 and Supplementary Figure 1?

Response: Please note that PKH26 are internalised over a long period of incubation time and some of the apoptotic stimulus as shown in Supplementary Figure 2 require >10 hrs of incubation. Thus, we chose not to use PKH26 in these experiments and AV staining is sufficient to visualise the formation of FOOD and F-ApoEVs.

7. Define all abbreviations in the legends (CLSM, ESID, SEM, etc.). They are defined in the text, but not in the respective legend.

Response: We have made these changes accordingly.

Reviewer #2 (Remarks to the Author):

The authors have successfully answered all points I raised in the previous round. There are only few minor issues.

1. *(Lines 102-106, Supplementary figure 3): Please state what is PANX1, and explain the relevance of the PANX1 cleavage for the data shown.*

Response: We have revised the manuscript appropriately to explain the relevance of the PANX1 cleavage.

2. *(Supplementary figure 7, legend): Words “nucleus” and “exposed phosphatidylserine” should be interchanged.*

Response: Thanks for picking up this typo. The Figure legend has been revised accordingly.

3. *(Lines 550-554, Supplementary Figure 12): Please provide more information the functional enrichment analysis. This includes explaining what are the p-values shown on this figure, and stating the statistical hypothesis tested and the parameters of the distribution.*

Response: We have revised the manuscript and figure legend appropriately to add additional information explaining that “the statistical significance for plots generated through enrichment analyses were calculated by hypergeometric uncorrected p-value” as determined by Funrich software.

4. *Spelling errors that changed the meaning of words: “thought” (line 284), “nothing” (line 331).*

Response: These spelling errors has now been corrected.